# Record thermopower found in an IrMn-based spintronic stack

Sa Tu[1,17], Timothy Ziman [2,3,4,17], Guoqiang Yu [5,6,17], Caihua Wan[6,17], Junfeng Hu [1,7,17], Hao Wu [5,17], Hanchen Wang [1], Mengchao Liu[8], Chuanpu Liu[1], Chenyang Guo[6], Jianyu Zhang[1], Marco A. Cabero Z. [9,1], Youguang Zhang[1], Peng Gao [8,10,11], Song Liu[9], Dapeng Yu[9,8], Xiufeng Han [6], Ingrid Hallsteinsen[12,13], Dustin A. Gilbert [14], Peter Wölfle[15], Kang L. Wang [5], Jean-Philippe Ansermet[7], Sadamichi Maekawa[16,4] & Haiming Yu [1✉]

The Seebeck effect converts thermal gradients into electricity. As an approach to power technologies in the current Internet-of-Things era, on-chip energy harvesting is highly attractive, and to be effective, demands thin film materials with large Seebeck coefficients. In spintronics, the antiferromagnetic metal IrMn has been used as the pinning layer in magnetic tunnel junctions that form building blocks for magnetic random access memories and magnetic sensors. Spin pumping experiments revealed that IrMn Néel temperature is thickness-dependent and approaches room temperature when the layer is thin. Here, we report that the Seebeck coefficient is maximum at the Néel temperature of IrMn of 0.6 to 4.0 nm in thickness in IrMn-based half magnetic tunnel junctions. We obtain a record Seebeck coefficient 390 (±10) μV K$^{-1}$ at room temperature. Our results demonstrate that IrMn-based magnetic devices could harvest the heat dissipation for magnetic sensors, thus contributing to the Power-of-Things paradigm.

[1] Fert Beijing Institute, BDBC, School of Microelectronics, Beihang University, 100191 Beijing, China. [2] Institut Laue-Langevin, 38042 Grenoble, France. [3] Université de Grenobles-Alpes, and CNRS, LPMMC, 38042 Grenoble, France. [4] Kavli Institute for Theoretical Sciences, University of Chinese Academy of Sciences, 100190 Beijing, China. [5] Department of Electrical Engineering, University of California, Los Angeles, CA 90095, USA. [6] Beijing National Laboratory for Condensed Matter Physics, Institute of Physics, University of Chinese Academy of Sciences, Chinese Academy of Sciences, 100190 Beijing, China. [7] Institute of Physics, Ecole Polytechnique Fédérale de Lausanne (EPFL), 1015 Lausanne, Switzerland. [8] Electron Microscopy Laboratory, School of Physics, Peking University, 100871 Beijing, China. [9] Shenzhen Institute for Quantum Science and Engineering (SIQSE), and Department of Physics, Southern University of Science and Technology (SUSTech), 518055 Shenzhen, China. [10] International Center for Quantum Materials, School of Physics, Peking University, 100871 Beijing, China. [11] Collaborative Innovation Center of Quantum Matter, 100871 Beijing, China. [12] Department of Electronic Systems, Norwegian University of Science and Technology, Trondheim 7491, Norway. [13] Advanced Light Source, Lawrence Berkeley National Laboratory, Berkeley, CA 94720, USA. [14] Material Science and Engineering Department, University of Tennessee, Knoxville, TN 37996, USA. [15] Institute for Theory of Condensed Matter, Karlsruhe Institute of Technology, 76049 Karlsruhe, Germany. [16] RIKEN Center for Emergent Matter Science (CEMS), Wako 351-0198, Japan. [17] These authors contributed equally: Sa Tu, Timothy Ziman, Guoqiang Yu, Caihua Wan, Junfeng Hu, Hao Wu. ✉email: haiming.yu@buaa.edu.cn

With the burgeoning Internet-of-Things (IoT)[1], consisting of billions of interconnected devices, a challenge has emerged in energizing these technologies using conventional batteries as power sources. This challenge motivates the crucial need for energy harvesting even at the level of nanodevices; on-chip thermoelectric devices[2] are proposed as one approach to respond to this need. However, to achieve an effective, deployable on-chip thermoelectric energy harvester[3], materials with a large thermoelectric power, i.e., Seebeck coefficient, are key. Spin caloritronics[4–8] is an interdisciplinary field which merges spintronics[9–11] with thermoelectrics[12–19] and has attracted tremendous attentions lately. A key device within spintronics and the IoT, and thus an attractive target to consider for spin caloritronics, is the magnetic tunnel junction (MTJ)[20–23]. The MTJ typically consists two ferromagnetic (FM) layers, separated by a thin insulating layer, with one of the magnetic layers pinned by exchange bias to an antiferromagnet, with IrMn being a popular choice. Recent works have investigated antiferromagnetic IrMn thin films using spin pumping[24,25], and report that its critical temperature depends on the film thickness[26]. This effect has also been reported in other antiferromagnets and is attributed to volume-based anisotropy energies[27]. The highly localized magnetic behavior of antiferromagnets near their critical temperature[28–32] gives rise to exciting physical phenomena. Spin colossal magnetoresistance has been reported near the critical temperature in antiferromagnetic $Cr_2O_3$[33]. In IrMn–CoFeB bilayers, it has been shown that the strength of the exchange-biasing depends on thickness. This has been put to use to make a thermopile[34]. It was also found that magnon transport around the phase transition of antiferromagnetic NiO layer[35] facilitates magnon-mediated spin torque switching. Furthermore, antiferromagnet-based thermoelectrics[36–39] has seen a surge in interest lately. Here, we investigate the thermoelectric response of IrMn/CoFeB near the IrMn critical temperature. We report a record Seebeck coefficient of 390 (±10) μV K$^{-1}$ at room temperature and a strong temperature dependence around the phase transition of IrMn, which depends on thickness. The highest value was 1.1 ± 0.1 mV K$^{-1}$ at 270 K for a thickness of 2.5 nm of IrMn. In our micro-structured thermoelectric devices, our temperature-dependent measurements show a sharp peak of the thermopower around the critical temperature ($T_{crit}$) of the IrMn film. The peak position of the thermopower can be shifted by changing the IrMn thickness, and directly correlates with the $T_{crit}$ characterized by magnetic susceptibility and X-ray magnetic linear dichroism (XMLD) measurements. A theoretical discussion is made, considering the influence of spin fluctuations near the critical temperature. The high thermopower in the IrMn-based device demonstrated in this work is an important step toward fulfilling the needs of ultra-low-power IoT applications[40].

## Results

**Seebeck coefficients vs. IrMn thickness at room temperature.** Figure 1a shows an illustrative diagram of the measurement setup for the IrMn/CoFeB-based thin film device. The multilayer full stack is Si/SiO$_2$/Ta(5)/IrMn($t_{AFM}$)/CoFeB(0.9)/MgO(2)/Ta(2) with thicknesses in nanometers. A high angle annular dark field scanning transmission electron microscopy (HAADF STEM) image is shown in the inset of Fig. 2a and verifies the film thickness and their high quality. The energy dispersive X-ray spectroscopy (EDX) images of each elements (Ta, Mg, Fe, and Mn) are shown in Supplementary Note 1 and Supplementary Fig. 1 and verify the designed compositional distributions. The FM CoFeB possesses a strong perpendicular magnetic anisotropy[41,42] as indicated by the large out-of-plane remanence in the measured hysteresis loop (Supplementary Note 2 and

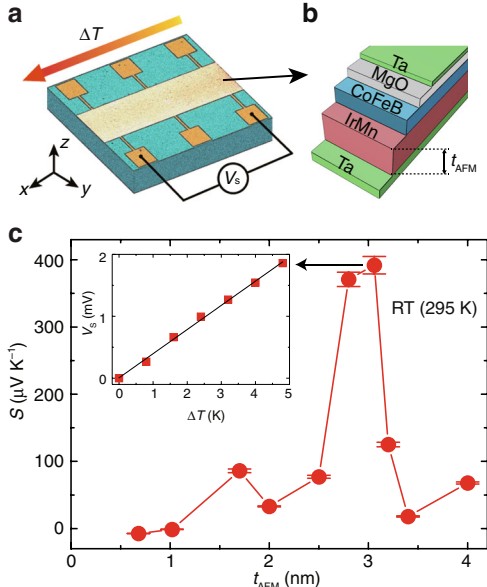

**Fig. 1 Enhanced Seebeck effect around the thickness-dependent Néel temperature. a** Schematics of Seebeck measurement on a rectangular bar (2 mm × 8 mm) of the IrMn-based multilayer. The temperature difference $\Delta T$ is applied in the plane of the thin film and mapped by a thermal camera (Supplementary Note 5 and Supplementary Fig. 4). **b** An illustration of the stack structure of the IrMn-based multilayer, where $t_{AFM}$ is the IrMn layer thickness. **c** Seebeck coefficients measured at room temperature (295 K) for a series of samples with $t_{AFM}$ between 0.6 and 4.0 nm. $t_{AFM}$ = 3.1 nm shows the largest observed Seebeck coefficient of 390 (±10) μV K$^{-1}$, which is extracted from the slope of the inset data showing a linear relation between the Seebeck voltage $V_S$ and the in-plane $\Delta T$. The error bars are taken from the linear fit to $V_S$ as a function of $\Delta T$ (e.g., black line in the inset).

Supplementary Fig. 2a). The details of the magnetic multilayer fabrication can be found in Methods. The Seebeck voltage $V_S$, as identified in the illustration in Fig. 1a, was measured for each of the Ta/IrMn/CoFeB/MgO/Ta multilayers, at a series of $\Delta T$, as shown in Fig. 1c inset for $t_{AFM}$ = 3.1 nm. The Seebeck coefficient is defined by the slope of the $V_S$ versus $\Delta T$ plot (Supplementary Note 3 and Supplementary Fig. 3). The apparent thickness dependence arises from the thickness dependence of the Néel temperature, as discussed below. The Seebeck coefficient $S$ is optimal at room temperature when the thickness is 3.1 nm. We find a record high value of $S$ = 390 (±10) μV K$^{-1}$. This value is substantially larger than most metallic structures and even larger than some well-known thermoelectric materials, such as Bismuth (−70 μV K$^{-1}$)[43] and Bi$_2$Te$_3$ (−160 μV K$^{-1}$)[44,45]. The figure of merit is estimated in Supplementary Note 4.

**Temperature dependence of thermopower in an IrMn-based thermopile.** Figure 2a shows a schematic of the thermopile patterned from the IrMn-based thin film. The IrMn-based thin film is first patterned into periodic nanowires about 800 nm wide, which are then connected in series by Au nanowires, patterned by e-beam lithography. In this arrangement, the potential gradients of each of the thermoelectric bars add up. A scanning electron microscope (SEM) image of the thermopile (partially) is shown in Fig. 2b. A Joule heater is integrated on-chip to generate an in-plane $\Delta T$ across the thermopile (Supplementary Note 6 and Supplementary Fig. 5). An AC current $I_{AC}$ is applied through the heating element at a frequency of 17 Hz and the Seebeck voltage is measured with a lock-in amplifier at twice the frequency based

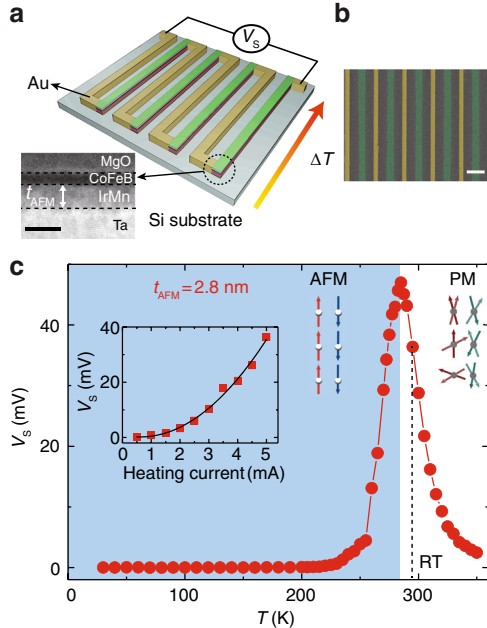

**Fig. 2 Temperature-dependent Seebeck effect measured on a meandering thermopile. a** An illustrative diagram of the thermopile consisting of periodic IrMn-based thin film bars connected in series with gold leads. The temperature difference $\Delta T$ is applied in-plane by a Joule heater (Supplementary Note 6 and Supplementary Fig. 5). The inset shows a high angle annular dark field scanning transmission electron microscopy (HAADF STEM) image of the IrMn-based magnetic multilayer, where $t_{AFM}$ is the IrMn layer thickness. The black scale bar is 5 nm. **b** A color-coded scanning electron microscope (SEM) image of the meandering thermopile. The green and yellow bars are the 800 nm wide IrMn-based thin films and the 400 nm wide gold leads. The white scale bar is 2 μm. **c** The Seebeck voltage measured on the sample with IrMn thickness $t_{AFM} = 2.8$ nm, from 10 to 350 K. The antiferromagnetic (AFM) phase in light blue and paramagnetic (PM) phase in white are divided by a critical temperature $T_{crit} \approx 285$ K. The dashed line indicates room temperature. Inset shows the Seebeck voltage measured at RT as a function of the heating current up to 5 mA. Red squares are measured data. The black line shows a quadratic fit to the data.

on the relation $V_{Seebeck} \propto \Delta T \propto I_{AC}^{2}$. An inherent advantage of such an integrated thermoelectric device[46] is that the temperature dependence of the Seebeck coefficient can easily be measured. Figure 2c shows the Seebeck voltage measured as a function of temperature on the sample with $t_{AFM} = 2.8$ nm, from 10 to 350 K. The results show that $V_S$ is vanishingly small at low temperatures (<250 K), then increases to a prominent peak of 47 mV at 285 K, then returns to almost zero (<5 mV) at high temperatures (>330 K). The $V_S$ at RT (295 K) is found to be ~36 mV. By varying the applied heating AC current, the thermopower at RT (295 K) shows a clear quadratic dependence (Fig. 2c) due to the Joule-heating-induced second harmonic response (See "Methods" for the details of the measurement technique). More data with other samples is shown in the Supplementary Note 7 and Supplementary Fig. 6. In particular, we demonstrate that a thermo-electric voltage of 0.4 V can be obtained with only 20 repeats of IrMn-based multilayer and a Joule heating of about 20 mA (Supplementary Note 8 and Supplementary Fig. 7).

**Thickness dependence of the critical temperature and the XMLD results.** At room temperature, the samples with $t_{AFM}$ of 2.8 and 3.1 nm show the largest Seebeck coefficients of all measured films. If we consider the best performance at any

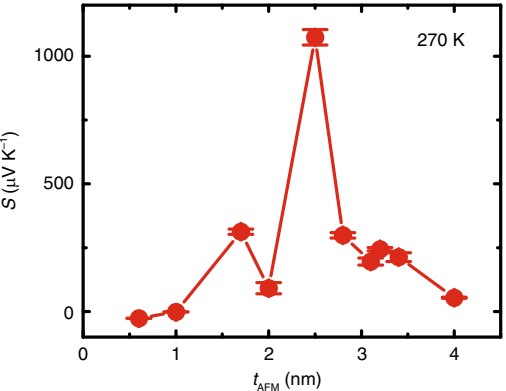

**Fig. 3 Thickness-dependent Seebeck effect at the temperature of 270 K.** The Seebeck coefficients at 270 K for samples with different thickness of 0.6, 1.0, 1.7, 2.0, 2.5, 2.8, 3.1, 3.2, 3.4, and 4.0 nm are extracted from the temperature dependence measurements. For the sample with $t_{AFM} = 2.5$ nm, the Seebeck coefficient peaks at 1.1 ± 0.1 mV K$^{-1}$. The error bars are extracted from the results difference of measurements which were repeated ten times.

temperature, then the Seebeck coefficients of the samples with $t_{AFM}$ of 2.0 and 2.5 nm were found to be much larger than these other two, at their critical temperatures ($T_{crit}$) of 260 and 270 K, respectively (Supplementary Note 9 and Supplementary Fig. 8). The data points in Fig. 3 are Seebeck coefficients at 270 K obtained from the temperature-dependent measurements on samples with different thicknesses (Supplementary Note 7 and Supplementary Fig. 6). The Seebeck coefficient peaks at the sample with $t_{AFM} = 2.5$ nm reaching 1.1 ± 0.1 mV K$^{-1}$ at 270 K. Hence, the Seebeck coefficient peak position depends jointly on the measurement temperature and the IrMn thickness. The reason for $S$ peaking at a specific couple of the temperature and thickness is twofold: (1) The temperature-dependent Seebeck coefficient peaks around the IrMn Néel temperature $T_{crit}$ (Fig. 2c); (2) $T_{crit}$ is strongly dependent on the thickness of IrMn[25].

For all samples, the $T_{crit}$ extracted from the peak positions of the $V_S$ versus $T$ plots (see Supplementary Note 7 and Supplementary Fig. 6) are plotted in Fig. 4a as a function of $t_{AFM}$. The black line in Fig. 4a is a fit based on equations in ref. [25] (fitting parameters can be found in Supplementary Note 11). In order to further confirm that the peak in the thermoelectric power indeed can be correlated to the critical temperature, the magnetic susceptibility of pure IrMn films—without a CoFeB layer—was measured as a function of temperature (Supplementary Note 10 and Supplementary Fig. 9). The $T_{crit}$ of pure IrMn samples with thicknesses of 2 and 4 nm are found to be around 290 and 310 K, respectively (blue triangles in Fig. 3a), which agrees reasonably well with the maxima in $V_S$ of samples with corresponding $t_{AFM}$. Samples with $t_{AFM}$ of 2.0, 3.2, and 4.0 nm are also characterized by XMLD. The XMLD signals arise from asymmetries in the orbital polarization in the plane of the film parallel and orthogonal to the cooling field; these two directions become equally occupied above the critical temperature and the XMLD signal decreases to zero. The XMLD intensity, shown in Fig. 4b, is fitted to a power-law expression which shows that the signal goes to zero at 266 and 290 K for $t_{AFM} = 3.2$ and 4.0 nm, respectively (see Supplementary Note 12 and Supplementary Fig. 10 for details). Disappearance of the XMLD signal indicates that the in-plane orbitals are equally occupied[47,48], consistent with a loss of long-range antiferromagnetic ordering. This is in good agreement with the results shown in Fig. 3a and previous results[25]. Recently, a drastic decrease of the spin Hall angle[49] was

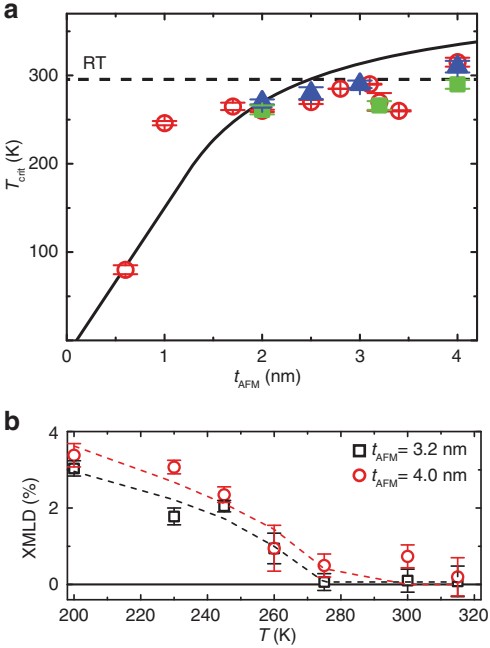

**Fig. 4 The critical temperatures $T_{crit}$ as a function of the IrMn layer thickness $t_{AFM}$. a** Theoretical prediction according to ref. [25] (black line), data from Seebeck measurements (Supplementary Note 7 and Supplementary Fig. 6) (red circles), susceptibility data (Supplementary Note 10 and Supplementary Fig. 9) (blue triangles), XMLD data (green squares). **b** Temperature dependence of the XMLD signals on the sample with $t_{AFM} = 3.2$ nm (black open squares) and the sample with $t_{AFM} = 4.0$ nm (red circles) determined by twice-integrating the XMLD signal under the $L_3$ peak. The error bars are generated from the standard deviation of the integrated signal. The solid line is a fit to a generic power-law expression.

observed in the antiferromagnet IrMn below 3 nm, which may be an independent evidence on the phase transition.

**Theoretical analysis and discussions.** We now turn to a theoretical interpretation of the critical temperature dependence of the thermopower, considering the coupling of magnetic fluctuations to transport close to the critical point of a metallic antiferromagnet. The Seebeck coefficient for weakly correlated conduction electrons in a metal can be written in the form

$$S = \frac{\pi^2 k_B^2 T}{3q} \frac{\sigma'(E_F)}{\sigma(E_F)}, \quad (1)$$

here $q$ is the charge of the carriers, and $\sigma(E) = \frac{1}{3} e^2 D(E) \rho(E)$ the energy dependent conductivity, which is a product of the density of states $\rho$ and a diffusivity $D(E) = v_F(E)^2 \tau(E)$, which itself depends on the Fermi velocity ($v_F$) and a scattering time ($\tau$). To display a strong temperature dependence one or several of these quantities must vary strongly with $T$. Non-monotonicity must appear because of rapid temperature variation in the factor

$$\frac{\sigma'(E_F)}{\sigma(E_F)} = 2 \frac{\partial \ln v_F(E)}{\partial E}\bigg|_{E=E_F} + \frac{\partial \ln \tau(E)}{\partial E}\bigg|_{E=E_F} + \frac{\partial \ln \rho(E)}{\partial E}\bigg|_{E=E_F}. \quad (2)$$

At finite temperatures these equations need to be generalized to include the effect of smearing over energies with the Fermi–Dirac distribution, but this will give a smooth change. To explain a strong temperature dependence, we can invoke either a temperature-dependent nesting that could alter the density of states and Fermi velocity, or a strong variation of the scattering

time from, for example, spin fluctuations. These will be greatest close to the magnetic transition. Experimentally, chromium metal[50,51] has a maximum in the thermopower close to (but just below) the Néel temperature, as is also seen in the antiferromagnetic pnictide $EuFe_2As_2$[52]. The compound $Mn_3Si$[53] also shows an enhancement in absolute value of the thermopower, but its maximum is above the critical ordering temperature. Thus, it appears to be general to have an enhanced thermopower in the critical region for itinerant antiferromagnets, but not necessarily centered exactly on the critical temperature. What is surprising, and interesting, in the current measurements is that the enhancement seen is much sharper than in such bulk materials. The most likely clue to the explanation is the fact that the room temperature critical temperature is well below the bulk Néel temperature (700 K for bulk IrMn[25]). This is most likely due to dimensional crossover from three to two-dimensional fluctuations and thus we should be in a strongly fluctuating regime. Very recently there has appeared an explicit theory[54], predicting a strong enhancement in the thermopower coming from an increase in derivative of the scattering rate at the Fermi energy $\epsilon_F$, resulting in a Seebeck coefficient that increases with a power law $t^{-\alpha}$ on the reduced temperature $t = \left(\frac{T - T_N}{T_N}\right)$ up to a maximum determined by the small cutoff $t \geq \frac{T_N}{\epsilon_F}$. The power law divergence $\alpha$ is particularly strong in two dimensions, $\alpha = 2$ compared to the bulk value of 3 and, in addition, the decreasing Néel temperature from dimensional reduction leads to a smaller cutoff. This theory is based on coupling of the self-energy of the electrons to the critical spin fluctuations close to a quantum critical point and includes loss of momentum conservation coming from impurity scattering. Thus, whether the theory is simply applicable here or not, it gives weight to the argument that the decrease in Néel temperature with crossover to low-dimensional fluctuations leads to the enhanced anomaly in Seebeck coefficients in our films compared to past measurements on bulk samples. More details on these theoretical considerations[55–59] are provided in Supplementary Note 13.

**The influence of the CoFeB layer on the Seebeck coefficient.** The IrMn phase transition has been shown to be responsible for the large Seebeck coefficient of the IrMn-based multilayer. If IrMn layer is removed from the multilayer stack, the Seebeck coefficient drops drastically to merely about 1% (Ta/CoFeB/MgO/Ta in Fig. 5) lower than that of $Ni_{81}Fe_{19}$[5], which proves again that the IrMn phase transition is the key ingredient for achieving the large Seebeck coefficient at room temperature. We now investigate the role of the CoFeB layer in the observed large Seebeck coefficient in the IrMn-based half MTJ. A control sample withdrawing the CoFeB layer (Ta/IrMn(3.1)/MgO/Ta in Fig. 5) is measured and a Seebeck coefficient of 82 ($\pm3$) μV K$^{-1}$ is found at room temperature, which is considerably reduced in comparison with the full stack (Ta/IrMn(3.1)/CoFeB/MgO/Ta in Fig. 5) but still larger than other antiferromagnets such as Cr around phase transition[13]. The temperature-dependent thermopower is measured for both samples with and without CoFeB layer and is found to exhibit a peak around approximately the same temperature (≈290 K), as shown in Supplementary Note 14 and Supplementary Fig. 11, which suggests that the CoFeB layer does not influence much the phase transition of IrMn.

A recent work[60] claims that the Dzyaloshinskii–Moriya interaction exists in IrMn/CoFeB/MgO thin film indicating the Rashba effect may play a role. The Seebeck coefficient may be influenced by the CoFeB layer due to a Rashba-induced effective electric field $E_R$ that originates from the asymmetric electron potential across the IrMn/CoFeB interface[61], which

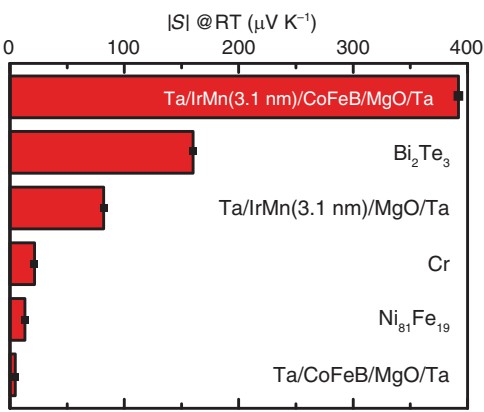

**Fig. 5 Seebeck coefficients of various materials.** The Seebeck coefficient of IrMn-based magnetic multilayer in comparison with conventional thermoelectric materials and magnetic materials in absolute value at room temperature. The values for $Bi_2Te_3$, Cr and $Ni_{81}Fe_{19}$ are taken from refs. [5,13,44] respectively. A direct comparison of the Seebeck voltage measured as a function of temperature on multilayer samples with and without CoFeB layer is shown in Supplementary Note 14 and Supplementary Fig. 11.

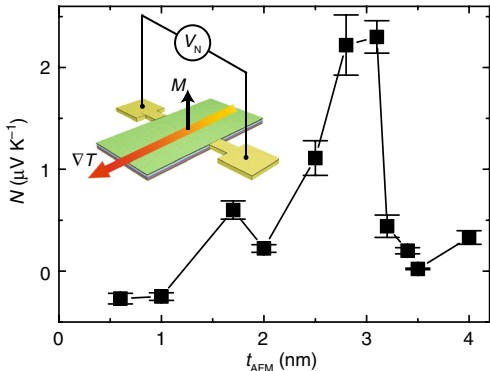

**Fig. 6 Anomalous Nernst effect.** The anomalous Nernst coefficients measured as a function of the IrMn thickness at room temperature. The error bars indicate the signal-to-noise level of the anomalous Nernst measurements. The inset illustrates the anomalous Nernst effect with Nernst voltage $V_N$ measured normal to temperature gradient $\nabla T$ with perpendicular magnetization $M$. The error bars are extracted from the results difference of measurements which were repeated ten times.

could possibly shift up or down the Fermi level $E_F$ and consequently affect all three energy derivative terms[62] at the Fermi level in Eq. (2). This could be comparable to tuning of the Fermi level by an external gate-generated electric field $E_g$. We note that it has been proven that gating can dramatically modulate the Seebeck coefficients, e.g., in two-dimensional materials such as graphene[63,64]. The CoFeB layer may also induce the magnetic proximity effect[65], which generates an effective internal magnetic field in the adjacent IrMn layer. This effective magnetic field may influence the spin fluctuations at the IrMn/CoFeB interface and eventually enhance the Seebeck coefficient. Furthermore, the magnon spin current at the interface of IrMn/CoFeB can contribute to the spin Seebeck effect as studied previously[28,29,31,33,66,67], but the electric voltage due to this effect is perpendicular to the temperature gradient and typically three orders of magnitude smaller than the Seebeck voltage detected in this work.

**Anomalous Nernst effect as a function of IrMn thickness.** In addition to the Seebeck effect, the anomalous Nernst effect (ANE)[68–73] is also investigated with different IrMn thicknesses with magnetic fields applied out of the plane of the film and ANE voltages measured along a line normal to the in-plane temperature gradient. The Nernst coefficients at room temperature are extracted for all samples (see Supplementary Note 15 and Supplementary Fig. 12), and exhibit comparable thickness dependence to that of the Seebeck coefficients. The sample with $t_{AFM} = 3.1$ nm shows the largest Nernst coefficient up to 2.3 (±0.2) $\mu V\,K^{-1}$ (Fig. 6). The thickness dependence of the Nernst effect parallels that of the Seebeck effect (Fig. 1). That the two are correlated implies that the ferromagnet/antiferromagnet interface plays an important role not only in the Seebeck effect but also in the ANE. The influence of the CoFeB magnetization on the Seebeck coefficient[6,7] is found to be rather insignificant (≈0.5%) as demonstrated by measuring $S$ in an external field applied normal to the film plane (Supplementary Note 16 and Supplementary Fig. 13) strong enough to flip the magnetization. These measurements show a hysteresis loop similar to the ANE measurement, with switching fields that agree well with magnetization measurements (Supplementary Note 2 and Supplementary Fig. 2). The exchange bias appears to be rather small (around

1 Oe), which is due to the thicknesses used in this work are smaller than the minimal thickness (≈10 nm) needed for exchange bias as used in ref. [34].

## Discussion

In summary, we report record Seebeck coefficients of 390 $\mu V\,K^{-1}$ in IrMn-based multilayer thin films at room temperature. Temperature-dependent measurements of the thermopower exhibits a prominent peak near the critical temperature of the IrMn. The peak position tracks the critical temperature of IrMn, which is found to strongly depend on the IrMn thickness, as was reported previously in spin pumping experiments[24]. The strong temperature dependence of the Seebeck coefficient is supported by a theory based on spin fluctuations around the critical phase transition temperature. Our results demonstrate a simple method to achieve large thermopower in magnetic multilayers by tuning the thickness of the antiferromagnetic IrMn layer. In a spintronic device, the different thicknesses of IrMn required to achieve the thermoelectric and magnetic pinning properties at a given temperature point can be solved by integrating two IrMn layers in the stack. Devices predicated on these results present great opportunities for on-chip energy harvesting for future Power-of-Things applications.

## Methods

**Sample fabrication.** The magnetic thin films investigated in the experiments consisting of $SiO_2$(sub)/Ta(5)/$Ir_{22}Mn_{78}$(x)/$Co_{20}Fe_{60}B_{20}$(0.9)/MgO(2)/Ta(2) (thicknesses in nanometers) were grown by magnetron sputtering at room temperature. The bottom 5 nm Ta layer is used as adhesion layer to the substrate. The top 2 nm Ta is deposited as a capping layer to prevent oxidation during the annealing process and patterning. The Silicon substrate was capped with 100 nm $SiO_2$. The chamber base pressure was lower than $2 \times 10^{-8}$ Torr. After growth the samples were annealed at 250 °C with an out-of-plane field. The sample was subsequently patterned into a meandering. The periodic IrMn/CoFeB/MgO stripes (12 μm in width, 1500 μm in length, and nearly 10 nm in thickness) were created by using optical lithography and ion beam etching. The periodic Au stripes (10 μm in width, 1500 μm in length, and 120 nm in thickness) were prepared by optical lithography and electron beam evaporation to connect with the IrMn/CoFeB/MgO stripes. The meandering Au stripes (4 μm in width, 2000 μm in length, and 120 nm thick) were patterned to form a Joule heater.

**Sample characterization.** The HAADF STEM images and the corresponding EDX mappings were acquired at an aberration-corrected FEI (Titan Cubed Themis G2) operated at 300 kV equipped with an X-FEG gun and Bruker Super-X EDX detectors.

**Thermoelectric measurement protocol**. The thermoelectric measurements on the IrMn-based integrated meandering device were carried out in a Physical Property Measurement System (PPMS). A Keithley 6221 current source and a lock-in amplifier SR830 were used in the Seebeck voltage measurement. The measurement details are shown below: a 5 mA AC current at a frequency of 17 Hz was applied to the gold Joule heater. Considering that the Joule heating power is proportional to the square of the heating current, the Seebeck voltage was measured based on the second harmonic mode of the lock-in amplifier. The isothermal measurement was conducted from 10 to 350 K using PPMS.

**XMLD measurements**. The sample was field cooled from 380 K with a 0.5 T in-plane field to 200 K, freezing in a particular antiferromagnetic alignment. Then, X-ray absorption/linear dichroism (XAS/XMLD) measurements were performed in a normal geometry, with the signal being measured from the total electron yield (TEY), capturing the Mn $L_{2,3}$ edges. Measurements were taken at increasing temperatures between 200 and 320 K. These measurements were performed at the Advanced Light Source on BL 4.0.2.

## Data availability

The authors declare that the main data supporting the findings of this study are available within the article and its Supplementary Information files. Extra data are available from the corresponding authors upon reasonable request.

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

## Acknowledgements
We thank S. Granville, M. Matsuo, J. Ieda, Y. Ohnuma, Z. Ma, M. Abid, A. Hamzić, and A. Fert for helpful discussions. We wish to acknowledge the support by NSF China under Grant nos. 11674020, U1801661, and 111 talent program B16001. This project was supported by the National Key Research and Development Program of China under Grant nos. 2016YFA0300802 and 2017YFA0206200. This work was also financially supported by the Key R&D Program of Guangdong Province (2018B030326001), the Guangdong Innovative and Entrepreneurial Research Team Program (2016ZT06D348), the Science, Technology and Innovation Commission of Shenzhen Municipality (no. ZDSYS20170303165926217). We also would like to acknowledge the support by the Sino-Swiss Science and Technology Cooperation (SSSTC, Grant No. EG 01-122016) for J. H. We gratefully acknowledge Electron Microscopy Laboratory in Peking University for the use of Cs corrected electron microscope. This work is financially supported by ERATO-JST (JPMJER1402), and KAKENHI (nos. 26103005, JP16H04023, and JP26247063) from MEXT, Japan. This research used resources of the Advanced Light Source, which is a DOE Office of Science User Facility under contract no. DE-AC02-05CH11231.

## Author contributions
J-P.A., S.M., and H.Y. conceived and designed the experiments. G.Y., H.W., and K.L.W. prepared the IrMn-based multilayer samples. M.L., P.G., and D.Y. conducted the TEM characterization. S.T., H-C.W., J.Z., C.L., Y.Z., and H.Y. designed and fabricated the thermoelectric devices. S.T., C.W., J.H., C.G., X.H., H-C.W., M.A.C.Z., S.L., and H.Y. performed the thermoelectric measurements. S.T., J.H., H-C.W., and H.Y. analyzed the data. S.M., T.Z., S.T., J-P.A., and H.Y. developed the explanation of the experiments. T.Z., P.W., and S.M. provided the theory. J-P.A. and H.Y. supervised the experimental study. I.H. and D.A.G. conducted the XMLD measurements and analysed the data. S.T., H-C.W., D.A.G., T.Z., S.M., and H.Y. wrote the paper and the supplementary information.

## Competing interests
The authors declare no competing interests.

## Additional information

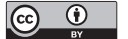

