## [Peer Review File · Nature Communications]

Reviewers' comments:

Reviewer #1 (Remarks to the Author):

Tu et al. present thermoelectric measurements of an IrMn/CoFeB half MTJ system which demonstrates a very large Seebeck coefficient in comparison to other measurements of systems within the field. The demonstration of the enhancement of the size of the Seebeck coefficient upon the antiferromagnet-paramagnet transition in IrMn is demonstrated in 2 ways, adding validity to the argument. On the whole the work is of a good scientific standard and presents a clear and elegant explanation for the results found. The field of self-powering devices is clearly very topical at the moment and the work does a good job in providing the necessary context in that regard. However, in its current form I do not believe it is suitable for publication in Nature Communications for the following reasons:

- No thorough error analysis accompanies any of the measurements presented in this work apart from those attached to the XMLD measurements. If the authors claim that they have achieved a record thermopower for a device, then such a claim needs to be held up to a proper scrutiny, which requires the inclusion of the relevant errors associated with such a measurement. I also believe that the inclusion of error bars would add context to the scale of the achievement by allowing for statistical significance tests against the current state of the art within the field, which would make a stronger case for publication. If the error bars are too small to see on a figure then please say so, but an error bar accompanying the presented values of the Seebeck coefficient is essential.
- A proper sense of the scale of the achievement is lacking throughout the work. I say this because none of the figures provide any real comparison against the state of the art, which for a claim of a record thermopower, I think would be essential. The addition of a summary of the thermopower for various devices types or simply a data set on a similar system where the IrMn is replaced by a metal with a reasonably high thermopower would be useful in this regard. The difference between the two in the size of the Seebeck coefficients should be clear and this would illustrate the scale of the achievement in a more striking way than current visualization of the data.
- I believe the section regarding the XMLD measurements could do with being revisited for a number of reasons:
 - 1) The figures in the supplementary material are difficult to interpret due to the small size in the signal. As they are at the moment, I can see no clear evidence for any XMLD in figure 12 of the supplementary information. Please replot the data here as the XAS scans of the two orientations of the field relative to the sample and then the resultant XMLD on a much larger scale to make it clearer.
 - 2) A full and comprehensive description of the physical process used in these XMLD measurements such as: how were the two orientations measured and which energy was selected for the normalization of the scans should be included. I include this comment as the scan presented for the 4 nm IrMn sample in Fig 12 of the supplementary material looks like noise or that it hasn't been normalized properly. A proper description of the method would make it easier to tell whether this is the case or not.
 - 3) It is also not clear as to where the T_B introduced in line 156 originates from? Presumably this is the blocking temperature, though isn't explicitly listed in the work. Is this supposed to be T_C or T_B ? If it is T_B I would like to see a comprehensive description of why this is the case and why it's not T_C ? I would not expect the XMLD signal to be sensitive only to T_B and not T_C . It is written as T_C in the supplementary information so perhaps this is a typo? Regardless, it should be clarified in future revisions of the manuscript.
- I also think that there are some physical processes mentioned in the paper that could do with some further explanation such as: the anomalous Nernst effect and its relationship to the Seebeck effect? This is included in the supplementary material but would benefit from having a sentence or two in the main text. The explanation included for the XMLD interpretation in the methods section should be moved to the main document.

A few minor suggestions for improvements would be:

- If the condition listed on the right-most constituent of equation 2 applies to all elements of the equation then please put it there for all. If not, please specify this in the accompanying text.
- The nature of the phase transition being probed in the system should be listed explicitly in the introductory paragraph. There are a number of systems that undergo phase transitions either into or out of antiferromagnetic states and saying 'boosted around the antiferromagnetic phase transition' doesn't provide the necessary specificity.
- I think that the data presented in table 1 would be better demonstrated in graphical form, with the IrMn thickness as the abscissa axis and T_{Crit} , S (RT), $S(T_{\text{Crit}})$ and N (RT) as the ordinate axes. This would be a representation of the data that is easier to digest than the table. Error bars should be included for all measurements here.
- The explanation of the data using equation 2 is clear and elegant, but it is lacking in description and context. I would like to see some of the discussion included in the supplementary material moved to the main document to rectify this.

Overall, I think that the work presented in this manuscript presents a clear step forward towards the realisation of self-powered devices with the order of magnitude of the Seebeck coefficient seen at the phase transition in the AF layer. The manuscript would benefit from making this point more evident by comparing it to the current state of the art more explicitly, most preferably in graphical form. The reason for this enhanced value is explained clearly and I particularly enjoyed the use of equation 2 in helping facilitate the explanation. I believe that if the authors were able to address the points raised in this review that the manuscript may be suitable for publication in Nature Communications after a further review of the revised manuscript.

Reviewer #2 (Remarks to the Author):

The paper discusses measurements of the Seebeck voltage at Ta/MgO/CoFeB/IrMn/Ta thin film structures that are also used in, e.g., magnetic tunnel junctions. The authors show, that the Seebeck voltage V_s has pronounced maxima at a temperature that roughly corresponds to a transition from antiferromagnetic to paramagnetic of the IrMn film.

The values for the Seebeck coefficients S reach about 1 mV / K, which is an extraordinarily large value for metallic systems. While this is of large interest, the manuscript in its present form does not meet the criteria of Nat. Com. I suggest to address the following issues:

1. As seen from the older references given below, a maximum of the Seebeck coefficient at the transition temperature of antiferromagnets was found already in the 1970-ies theoretically and experimentally. Thus the fact, that such a maximum is present in IrMn is nothing special. The important and strong point of the manuscript is the magnitude of S , which is much larger than reported for other antiferromagnets. The theoretical explanation offered is, however, true for any antiferromagnet. Thus I challenge the authors to search for an explanation of the large S , which is more specific for the IrMn and the comparatively complex film stack used by them. Additional measurements of, e.g., the electric resistivity vs temperature and the Seebeck coefficient of bare IrMn films could help to understand the effect and the role of the CoFeB and Ta better.
2. In addition, the results given for the anomalous Nernst effect are difficult to relate with the Seebeck voltage. While the main Seebeck voltage may be generated in the IrMn, the main contribution to the ANE comes probably from the CoFeB. Thus the contribution of the ANE and the measurements of the magneto-thermopower (supplement) to the understanding of the large S should be discussed in more detail.
3. To judge, how good the real potential of the devices is, the figure of merit ($zT=S/\rho_{\text{el}} \times \kappa_{\text{thermal}}^{-1}$) should be estimated (which includes an estimation of the electric resistivity ρ_{el} and the thermal conductivity κ_{thermal}).

4. The cited literature is somehow incomplete. The authors cite many paper from spin caloritronics, which has not much to do with their Seebeck measurements. The authors have probably overlooked some important papers from long ago such as:

M. Ausloos, Solid State Communications, Volume 21, Issue 4, January 1977, Pages 373-375

A. Fote et al., PHYSICAL REVIEW B VOLUME 8, NUMBER 5, PAGE 2099 (Sept. 1973)

Also younger papers such as H. Takaki et al., Appl. Phys. Lett. 110, 072107 (2017);

<https://doi.org/10.1063/1.4976574> are missing.

I cannot give a full list of adequate references, but the authors should at least cite some work on thermoelectrics with antiferromagnets.

The paper thus reports a potentially very valuable result, but it needs improvements and some additional work to be acceptable for Nat.Com..

One minor issue is the following: The figures 2 and sup-7 show the Seebeck voltage of the meandering sample. Why is this voltage at maximum around 45 mV in figure 2 and 400 mV in figure sup-7 ?

Reviewer #3 (Remarks to the Author):

The authors present a detailed thermopower study of IrMn/CoFeB thermopiles, and report a peak in the Seebeck coefficient at what appears to be the critical temperature of the IrMn thin film. A record value is observed at that temperature for a specific thickness.

The results are overall rather interesting, and could be published in Nature Communications if several primary and secondary aspects are satisfactorily addressed.

Primary:

- Line 90 : " This value is substantially larger than most metallic structures and 91 even larger than some well-known thermoelectric materials, such as Bi₂Te₃ (-160 μ V K⁻¹)²¹." Please provide more refs/background to substantiate this claim.

- In Fig. 3, the link between the Seebeck coeff. Peak and the IrMn AFM critical temperature is experimentally tenuous at present, with only a few magnetic susceptibility and XLMD measurements to overlap with the temperature of the Seebeck coeff. Peak. Please perform additional magnetic susceptibility measurements at other thicknesses in order to conform to experiment and to the model (black line of Fig. 3)

- Line 194: "To fully understand the CoFeB layer influence requires 195 further investigation and is beyond the scope of this work." This record thermopower is found at the (supposed) IrMn critical temperature. In a commercial device, the IrMn would not be designed such that the device operates at the IrMn critical temperature, because doing so would not allow for magnetic pinning of one of the FM layers. Thus, for the main claim of the paper to stand, this requires that this IrMn layer not play a role in the magnetostatics of the film. This 1) weakens the importance of the paper's main claim and 2) requires that the IrMn be placed adjacent to non-magnetic layers so as to destabilize (electrically/magnetically) the system. Please discuss this in the context of Ref. 17, which associates the thermopower in these systems with exchange bias.

- The authors do not really discuss why the Seebeck coeff. Maximum is highest at a precise thickness. This will be required to publish in Nat. Comm.

- The theoretical description is rather vague.

Secondary

- Line 81 "with minimal intermixing": that doesn't seem so based on Fig. S1. Also, are each panel of Fig. S1 at the same position? One gets the impression that Fe and Mg (bright streaks) are right of Mn despite the total color scale map showing otherwise. Clearly, Fig. S1 is confusing as is.

Mechanics:

- Ordering of figure description out of order with Figure assembly. For example, it isn't natural to describe the record Seebeck value before describing the thermopile.
- Please check manuscript and SI consistency, e.g. : "Based on the single rectangular bar device in Supplementary Fig. 3", even though Fig S3 is about how to measure the thermal gradient using data extracted from the thermal camera.
- Please indicate IrMn thickness in all plots of all relevant Figures, including Fig. S4.
- Fig 1c: indicate the temperature in the graph
- The table caption is quite minimal. For example, is N the Nernst coefficient?? It isn't defined at the first appearance in the text?

English

Here are a few examples of minor English improvements that are needed throughout the text. Please improve throughout through a careful rereading.

Line 48: junctions - > junction

Line 58: We report a record Seebeck coefficient of up to $392 \mu\text{V K}^{-1}$, tunable over a wide 59 temperature range, including room temperature, by the thickness of the antiferromagnetic IrMn layer. -> improve English:

We report a record Seebeck coefficient of up to $392 \mu\text{V K}^{-1}$ that is tunable over a wide 59 temperature range, including room temperature, using the thickness of the antiferromagnetic IrMn layer.

"Line 62: "The peak position of the 62 thermopower"" -> you mean the temperature at which the peak occurs, please improve.

Critical temperature of IrMn : AFM?

SI: "which includes only 2 atoms of IrMn" -> atomic planes

"which had the highest Seebeck coefficient"

"The discrepancy may be because of different IrMn targets and crystal orientation of the thin film."

Line 86 : "The Seebeck coefficient S , is"

Response to referee reports

First of all, we thank all three reviewers for their time and efforts and the detailed review comments and suggestions which we find very constructive and helpful. We have taken them fully into account in our revised paper. We truly hope that you will find it now worthy of publication in Nature Communications.

Below we provide our point-to-point responses to all of the review comments.

Reviewer #1:

Tu et al. present thermoelectric measurements of an IrMn/CoFeB half MTJ system which demonstrates a very large Seebeck coefficient in comparison to other measurements of systems within the field. The demonstration of the enhancement of the size of the Seebeck coefficient upon the antiferromagnet-paramagnet transition in IrMn is demonstrated in 2 ways, adding validity to the argument. On the whole the work is of a good scientific standard and presents a clear and elegant explanation for the results found. The field of self-powering devices is clearly very topical at the moment and the work does a good job in providing the necessary context in that regard.

Response:

We thank Reviewer #1 for summarize our work nicely, and in particular pointing out our using 2 ways to demonstrate the large Seebeck effect, which adds validity to the argument.

We greatly appreciate Reviewer #1's positive comments on our work to be of a good scientific standard and presents a clear and elegant explanation for the results found and acknowledge that the field of self-powering devices is clearly very topical and our work does a good job in providing the necessary context.

However, in its current form I do not believe it is suitable for publication in Nature Communications for the following reasons:

-No thorough error analysis accompanies any of the measurements presented in this work apart from those attached to the XMLD measurements. If the authors claim that they have achieved a record thermopower for a device, then such a claim needs to be held up to a proper scrutiny, which requires the inclusion of the relevant errors associated with such a measurement. I also believe that the inclusion of error bars would add context to the scale of the achievement by allowing for statistical significance tests against the current state of the art within the field, which would make a stronger case for publication. If the error bars are too small to see on a figure then please say so, but an error bar accompanying the presented values of the Seebeck coefficient is essential.

Response:

We thank Reviewer #1 for pointing out to the lack of error analysis.

We have now included the error bars in Fig. 1C, showing the thickness dependence of Seebeck coefficients. The error bars are extracted from the linear fit to the Seebeck voltage versus temperature difference. The error bars are in general small since the dependence appears to be quite linear, but are still sizable. For example, we now write in the main text

"We report a record Seebeck coefficient of $392 (\pm 13) \mu V K^{-1}$ at room temperature..."

One sentence has been added to the caption of Fig. 1 as

"The error bars are taken from the linear fit to V_S as a function of ΔT (e.g. black line in the inset)."

In addition, we have also added error bars to the measured Nernst coefficients, which is now Fig. 5, see comments to other referees, below.

- A proper sense of the scale of the achievement is lacking throughout the work. I say this because none of the figures provide any real comparison against the state of the art, which for a claim of a record thermopower, I think would be essential. The addition of a summary of the thermopower for various device types or simply a data set on a similar system where the IrMn is replaced by a metal with a reasonably high thermopower would be useful in this regard. The difference between the two in the size of the Seebeck coefficients should be clear and this would illustrate the scale of the achievement in a more striking way than current visualization of the data.

Response:

We thank Reviewer #1 to give suggestions of showing a direct comparison against the state of the art and a similar system without IrMn. Following this constructive advice, we have grown magnetic multilayers by withdrawing the IrMn layer and leaving Ta directly in contact with the CoFeB layer. The Seebeck coefficients of such samples are measured and found to be quite small $\sim 4.6 \mu\text{V K}^{-1}$.

We conclude our paper with a comparison with other well-known materials in thermoelectrics and spintronics, e.g. Bi_2Te_3 , Cr and $\text{Ni}_{81}\text{Fe}_{19}$. The performance of these various materials are shown in the newly added Fig. 4.

2) I believe the section regarding the XMLD measurements could do with being revisited for a number of reasons:

1) The figures in the supplementary material are difficult to interpret due to the small size in the signal. As they are at the moment, I can see no clear evidence for any XMLD in figure 12 of the supplementary information. Please replot the data here as the XAS scans of the two orientations of the field relative to the sample and then the resultant XMLD on a much larger scale to make it clearer.

Response:

We thank Reviewer #1 for his/her attention to the XMLD results. In the revised Supplementary Fig. 10, the XMLD results (re-analyzed as discussed below) are plotted at $10\times$ magnification to improve clarity. To simplify the results, we have removed the heat-map and now present representative panels for 200 K and 300 K data. These plots clearly show the XMLD signal at the L_3 edge at 200 K and much weaker or non-existent at 300 K; no signal is resolved in any of the samples at the L_2 edge for any temperature, likely due to the weaker intensity. Since the difference between the XAS signals measured with the two polarizations is $<1\%$, they appear indistinguishable when plotted together (included below). Plotting the sum and difference is a valid representation of the data and we believe most-accurately conveys the results. We have therefore updated the Supplementary Fig. 10 accordingly.

2) A full and comprehensive description of the physical process used in these XMLD measurements such as: how were the two orientations measured and which energy was selected for the normalization of the scans should be included. I include this comment as the scan presented for the 4 nm IrMn sample in Fig 12 of the supplementary material looks like noise or that it hasn't been normalized properly. A proper description of the method would make it easier to tell whether this is the case or not.

Response:

We thank Reviewer #1 for this helpful suggestion on the XMLD analysis. First, we note that the noise floor in the XMLD signal is ≈ 0.004 , as determined from the local RMS variation, much smaller than the proposed feature, and so the reviewer is presumed to be addressing drift and scaling issues. The XMLD signal is quite small – as is typical for metallic AFMs – and is made worse by the small thickness of the IrMn layer, buried under a 2.9 nm capping layer, and so the data requires diligent analysis to remove any sources of error. Despite these challenges, we achieve results that are consistent with previous reports of the critical temperature.

To reduce the effects of drift and other chronic noise sources, we performed twenty sequential measurements with alternating linear polarization (A/B/A/B/etc.). Each of these scans were individually corrected for vertical offsets and linear drift in intensity, normalized, then measurements with the same polarization were averaged. This raw data shows the expected negative/positive feature pair typically associated with XMLD. The data was then translated (in energy) to account for inaccuracies in the monochromator resolution, with a target to minimize the overall difference signal. The issue is that long-period drift persists in the data, and so minimization was achieved at too-large of an energy translation, resulting in the positive-only feature presented in Version 1 of the text.

The data has been re-analyzed, focusing on the difference signal under the L_3 peak and limiting the energy translation. The XMLD data now shows the expected negative/positive signal, located at approximately the correct energy value, reported in previous works. To determine the intensity of the signal, the area of the XMLD signal under the L_3 peak was integrated twice. The twice-integrated area was plotted versus temperature, resulting in the new Fig. 3b. We note that these values are slightly lower than the previous values (by <10 K), but remain consistent with the results. We have accordingly updated the caption of Fig. 3b as

“...determined by twice-integrating the XMLD signal under the L_3 peak. The error bars are generated from the standard deviation of the integrated signal. The solid line is a fit to a generic power-law expression.”

and enriched the discussion on XMLD in the main text as

“The XMLD signals arise from asymmetries in the orbital polarization in the plane of the film parallel and orthogonal to the cooling field; these two directions become equally occupied above the critical temperature and the XMLD signal decreases to zero. The XMLD intensity, shown in Fig. 3b, is fitted to a power-law expression which shows that the signal goes to zero at 266 K and 290 K for $t_{AFM} = 3.2$ nm and 4.0 nm, respectively (see Supplementary Fig. 10 for details). Disappearance of the XMLD signal indicates that the in-plane orbitals are equally occupied^{46,47}, consistent with a loss of long-range antiferromagnetic ordering. This is in good agreement with the results shown in Fig. 3a and previous results²⁴.”

3) It is also not clear as to where the T_B introduced in line 156 originates from? Presumably this is the blocking temperature, though isn't explicitly listed in the work. Is this supposed to be T_C or T_B ? If it is T_B I would like to see a comprehensive description of why this is the case and why its not T_C ? I would not expect the XMLD signal to be sensitive only to T_B and not T_C . It is written as T_C in the supplementary information so perhaps this is a typo? Regardless, it should be clarified in future revisions of the manuscript.

Response:

We thank Reviewer #1 for pointing this out. Indeed, this should be T_{crit} and we have now corrected it in the main text.

The reviewer may be interested to know that, above the critical temperature T_{crit} there may still exist local magnetic ordering, but no long-range correlation. Since the X-ray beam is 10 μm in diameter, above the critical temperature we capture the signal from many “domains” which are randomly oriented and flipping with time. The average signal from these randomly oriented domains will be independent of the X-ray polarization, and thus the XMLD signal is zero. Further increasing the temperature to the bulk Néel temperature would not result in recovery of the signal, the XMLD will remain zero for $T > T_{crit}$.

- I also think that there are some physical processes mentioned in the paper that could do with some further explanation such as: the anomalous Nernst effect and its relationship to the Seebeck effect? This is included in the supplementary material but would benefit from having a sentence or two in the main text. The explanation included for the XMLD interpretation in the methods section should be moved to the main document.

Response:

We thank Reviewer #1 for the suggestion to add some further explanation to the anomalous Nernst effect in relation to the Seebeck effect. This is indeed a very helpful comment and suggestion which would make the claim for the large Seebeck coefficient in a more scientifically sound manner and make it a stronger case for publication.

We have therefore added a new figure (Fig. 5) and the following paragraph:

“In addition to the Seebeck effect, the anomalous Nernst effect (ANE)⁶⁴⁻⁶⁹ is also investigated with different IrMn thicknesses with magnetic fields applied out of the plane of the film and ANE voltages measured along a line normal to the in-plane temperature gradient. The Nernst coefficients at room temperature are extracted for all samples (see Supplementary Fig. 12), and exhibit comparable thickness dependence to that of the Seebeck coefficients. The sample with $t_{AFM} = 3.1$ nm shows the largest Nernst coefficient up to $2.3 (\pm 0.2) \mu\text{V K}^{-1}$ (Fig. 5)...”

Also, following the constructive advice of Reviewer #1, we have now moved two sentences about the explanation of XMLD data from the method to the main text as:

“The XMLD signals arise from asymmetries in the orbital polarization in the plane of the film parallel and orthogonal to the cooling field; these two directions become equally occupied above the critical temperature and the XMLD signal decreases to zero.”

A few minor suggestions for improvements would be:

- If the condition listed on the right-most constituent of equation 2 applies to all elements of the equation then please put it there for all. If not, please specify this in the accompanying text.

Response:

Indeed, the condition applies to all three terms of equation 2. We thank Reviewer #1 for pointing out this ambiguity and have now added it for all three terms.

- The nature of the phase transition being probed in the system should be listed explicitly in the introductory paragraph. There are a number of systems that undergo phase transitions either into or out of antiferromagnetic states and saying ‘boosted around the antiferromagnetic phase transition’ doesn’t provide the necessary specificity.

Response:

We fully agree with Reviewer #1 that more emphasis on the phase transition and its critical temperature should be mentioned in the abstract and introduction sections.

In response to this comment, we have sharpened the abstract and introductory statements. In particular, we point to the work on spin pumping (now Ref. 24).

We have now added one sentence in the abstract

“Spin pumping experiments revealed that IrMn Néel temperature is thickness-dependent and approaches room temperature when the layer is thin.”

In the introduction, we add a few references on phase transition in other antiferromagnets as

Lin, W., Chen, K., Zhang, S. & Chien, C. L. Enhancement of thermally injected spin current through an antiferromagnetic insulator. Phys. Rev. Lett. 116, 186601 (2016).

Qiu, Z. et al. Spin-current probe for phase transition in an insulator. Nat. Commun. 7, 12670 (2016).

Saglam, H. et al. Spin transport through the metallic antiferromagnet FeMn. Phys. Rev. B 94, 140412(R) (2016).

Prakash, A., Brangham, J., Yang, F. & Heremans, J. P. Spin Seebeck effect through antiferromagnetic NiO. Phys. Rev. B 94, 014427 (2016).

Wang, Y. et al. Spin pumping during the antiferromagnetic-ferromagnetic phase transition of iron-rhodium. Nat. Commun. 11, 275 (2020).

and one sentence referring a very recent work concerning antiferromagnetic NiO undergoing phase transition

“It was also found that magnon transport around the phase transition of antiferromagnetic NiO layer³⁴ facilitates magnon-mediated spin torque switching.”

[34] *Wang, Y. et al. Magnetization switching by magnon-mediated spin torque through an antiferromagnetic insulator. Science 366, 1125-1128 (2019).*

- I think that the data presented in table 1 would be better demonstrated in graphical form, with the IrMn thickness as the abscissa axis and T_Crit, S (RT), S(T_Crit) and N (RT) as the ordinate axes.

This would be a representation of the data that is easier to digest than the table. Error bars should be included for all measurements here.

Response:

We thank Reviewer #1 for pointing this out and the helpful suggestion for the improvement of presentation of the data. In fact, the thickness dependence of T_{Crit} and S (RT) have already been presented in Fig. 3a and Fig. 1c. We therefore plot the Nernst data N (RT) as a function of thickness in the newly added Fig. 5 and put the $S(T_{\text{Crit}})$ versus thickness as Supplementary Fig. 8.

- The explanation of the data using equation 2 is clear and elegant, but it is lacking in description and context. I would like to see some of the discussion included in the supplementary material moved to the main document to rectify this.

Response:

We thank Reviewer #1 for the appreciation of our using equation 2. We have now moved some of the discussions from the supplementary information to the main text and further enriched the paragraph as

“...What is surprising, and interesting, in the current measurements is that the enhancement seen is much sharper than in such bulk materials. The most likely clue to the explanation is the fact that the room temperature critical temperature is well below the bulk Néel temperature (700 K for bulk IrMn^{24}). This is most likely due to dimensional crossover from three to two-dimensional fluctuations and thus we should be in a strongly fluctuating regime. Very recently there has appeared an explicit theory⁵³, predicting a strong enhancement in the thermopower coming from an increase in derivative of the scattering rate at the Fermi energy ϵ_F , resulting in a Seebeck coefficient that increases with a power law $t^{-\alpha}$ on the reduced temperature $t = \left(\frac{T-T_N}{T_N}\right)$ up to a maximum determined by the small cut-off $t \geq \frac{T_N}{\epsilon_F}$. The power law divergence α is particularly strong in two dimensions, $\alpha = 2$ compared to the bulk value of 3 and, in addition, the decreasing Néel temperature from dimensional reduction leads to a smaller cut-off. This theory is based on coupling of the self energy of the electrons to the critical spin fluctuations close to a quantum critical point and includes loss of momentum conservation coming from impurity scattering. Alongside the crossover in antiferromagnetic fluctuations, from the bulk to a film geometry for the antiferromagnetism, there may also be a crossover from metallic to localized behavior in the conducting electrons. Thus whether the theory is simply applicable here or not, it gives weight to the argument that the decrease in Néel temperature with crossover to low-dimensional fluctuations leads to the enhanced anomaly in Seebeck coefficients in our films compared to past measurements on bulk samples. More details on these theoretical considerations⁵⁴⁻⁵⁸ are provided in Supplementary Section XIII.”

Overall, I think that the work presented in this manuscript presents a clear step forward towards the realisation of self-powered devices with the order of magnitude of the Seebeck coefficient seen at the phase transition in the AF layer. The manuscript would benefit from making this point more evident by comparing it to the current state of the art more explicitly, most preferably in graphical form. The reason for this enhanced value is explained clearly and I particularly enjoyed the use of equation 2 in helping facilitate the explanation. I believe that if the authors were able to address the points raised in this review that the manuscript may be suitable for publication in Nature Communications after a further review of the revised manuscript.

Response:

We greatly appreciate Reviewer #1's full support. We are also very grateful for his/her helpful suggestions on how to improve the manuscript to be suitable for Nature Communications. We have followed his/her suggestions in full.

Reviewer #2:

The paper discusses measurements of the Seebeck voltage at Ta/MgO/CoFeB/IrMn/Ta thin film structures that are also used in, e.g., magnetic tunnel junctions. The authors show, that the Seebeck voltage V_S has pronounced maxima at a temperature that roughly corresponds to a transition from antiferromagnetic to paramagnetic of the IrMn film.

The values for the Seebeck coefficients S reach about 1 mV / K, which is an extraordinarily large value for metallic systems. While this is of large interest, the manuscript in its present form does not meet the criteria of Nat. Com. I suggest to address the following issues:

1) As seen from the older references given below, a maximum of the Seebeck coefficient at the transition temperature of antiferromagnets was found already in the 1970-ies theoretically and experimentally. Thus, the fact, that such a maximum is present in IrMn is nothing special. The important and strong point of the manuscript is the magnitude of S , which is much larger than reported for other antiferromagnets. The theoretical explanation offered is, however, true for any antiferromagnet. Thus, I challenge the authors to search for an explanation of the large S , which is more specific for the IrMn and the comparatively complex film stack used by them. Additional measurements of, e.g., the electric resistivity vs temperature and the Seebeck coefficient of bare IrMn films could help to understand the effect and the role of the CoFeB and Ta better.

Response:

Indeed, we agree with Reviewer #2 that the strong point of our manuscript is the magnitude of S , which is much larger than reported for other antiferromagnets, such as Cr studied by Fote et al. (1973) being $21.5 \mu\text{V K}^{-1}$ at room temperature. We have added this reference, all others of the referee's list plus one other connected to the recent simulation paper. Following the suggestion of this and Reviewer #1, we have created a list of materials in the newly added Fig. 4.

In particular, we thank Reviewer #2 for raising a very important question as to why the magnitude of S found in our work is so much larger. As suggested by the reviewer that the answer may lie in the IrMn itself, we therefore grew a control sample with the same IrMn-based multilayer stack except by withdrawing the CoFeB layer. We have then measured the temperature dependence of the thermopower and put together the results with and without CoFeB layer in the newly added Supplementary Fig. 11. Thus, we find that by removing CoFeB, the peak remains around the same temperature (~ 290 K), but the peak amplitude decreases dramatically. This check experiment clarifies the central role of the phase transition in IrMn. It shows also that the CoFeB does not affect much this phase transition.

Furthermore, we suggest that the interface gives rise to a Rashba effect that may cause the enhancement of the IrMn Seebeck peak. The electrical resistivity is measured as a function of the temperature and is now shown in the newly added Supplementary Fig. 2b.

Thanks to the helpful suggestion by Reviewer #2, we have now added one paragraph in the main text to discuss the influence of CoFeB layer as

“The influence of the CoFeB layer on the Seebeck coefficient. The IrMn phase transition has been shown to be responsible for the large Seebeck coefficient of the IrMn-based multilayer. If IrMn layer is removed from the multilayer stack, the Seebeck coefficient drops drastically to merely about 1% (Ta/CoFeB/MgO/Ta in Fig. 4) lower than that of $\text{Ni}_{81}\text{Fe}_{19}$ ^{Error! Reference source not found.}, which proves again that the IrMn phase transition is the key ingredient for achieving the large Seebeck coefficient at

room temperature. We now investigate the role of the CoFeB layer in the observed large Seebeck coefficient in the IrMn-based half MTJ. A control sample withdrawing the CoFeB layer (Ta/IrMn(3.1)/MgO/Ta in Fig. 4) is measured and a Seebeck coefficient of $82 \mu\text{V K}^{-1}$ is found at room temperature, which is considerably reduced in comparison with the full stack (Ta/IrMn(3.1)/CoFeB/MgO/Ta in Fig. 4) but still larger than other antiferromagnets such as Cr around phase transition^{Error! Reference source not found.}. The temperature dependent thermopower is measured for both samples with and without CoFeB layer and is found to exhibit a peak around approximately the same temperature ($\sim 290 \text{ K}$), as shown in Supplementary Fig. 11, which suggests that the CoFeB layer does not influence much the phase transition of IrMn.

A recent work⁵⁹ claims that the Dzyaloshinskii-Moriya interaction exists in IrMn/CoFeB/MgO thin film indicating the Rashba effect may play a role. The Seebeck coefficient may be influenced by the CoFeB layer due to a Rashba-induced effective electric field E_R that originates from the asymmetric electron potential across the IrMn/CoFeB interface⁶⁰, which could possibly shift up or down the Fermi level E_F and consequently affect all three energy derivative terms⁶¹ at the Fermi level in Eq. 2. This could be comparable to tuning of the Fermi level by an external gate-generated electric field E_g . We note that it has been proven that gating can dramatically modulate the Seebeck coefficients, e.g. in two-dimensional materials such as graphene^{62,63}.

2) In addition, the results given for the anomalous Nernst effect are difficult to relate with the Seebeck voltage. While the main Seebeck voltage may be generated in the IrMn, the main contribution to the ANE comes probably from the CoFeB. Thus, the contribution of the ANE and the measurements of the magneto-thermopower (supplement) to the understanding of the large S should be discussed in more detail.

Response:

This is definitely a very important point that we have unfortunately overlooked and taken for granted previously. We sincerely thank Reviewer #2 for reminding us to give more details about the Nernst effect.

In response to the reviewer, we have added Fig. 5. The thickness dependence of the Nernst coefficient strikingly parallels that of the Seebeck coefficient. In connection with this new figure, we have added the following text:

“The thickness dependence of the Nernst effect (Fig. 5) parallels that of the Seebeck effect (Fig. 1). That the two are correlated is consistent with the Mott formula (Eq. 1), where the Seebeck and Nernst coefficients are proportional to energy derivative of the longitudinal and transverse conductivity (σ'_{xx} and σ'_{xy}) at the Fermi level, respectively.”

The referee might be interested to know that the Nernst effect may be considerably enhanced by the presence of CoFeB/IrMn interface. The fact that the Nernst and the Seebeck effects are correlated stems from the Mott relations, as stated in the manuscript. In general, according to the Mott relation, the Seebeck coefficient S can be written as Eq. 1 in the main text, which can also be written as below:

$$S = S_{xx} = \frac{\sigma_{xx}\alpha_{xx} + \sigma_{xy}\alpha_{xy}}{\sigma_{xx}^2 + \sigma_{xy}^2} = \frac{\pi^2 k_B^2 T}{3q} \frac{\sigma_{xx}\sigma'_{xx}(E_F) + \sigma_{xy}\sigma'_{xy}(E_F)}{\sigma_{xx}^2 + \sigma_{xy}^2}$$

The anomalous Nernst coefficient N can be written in a similar way as:

$$N = S_{xy} = \frac{\sigma_{xx}\alpha_{xy} - \sigma_{xy}\alpha_{xx}}{\sigma_{xx}^2 + \sigma_{xy}^2} = \frac{\pi^2 k_B^2 T}{3q} \frac{\sigma_{xx}\sigma'_{xy}(E_F) - \sigma_{xy}\sigma'_{xx}(E_F)}{\sigma_{xx}^2 + \sigma_{xy}^2}$$

Thus, the anomalous Nernst coefficient is also dependent on the three parameters [density of states (ρ), Fermi velocity (v_F) and a scattering time (τ)] that influences the Seebeck coefficient. Indeed, we have found that the anomalous Nernst coefficient shows a comparable thickness dependence to that of the Seebeck coefficients.

3) To judge, how good the real potential of the devices is, the figure of merit ($zT = S/\rho_{el} \times \kappa_{thermal} T$) should be estimated (which includes an estimation of the electric resistivity ρ_{el} and the thermal conductivity $\kappa_{thermal}$).

Response:

We appreciate Reviewer #2's suggestion on the estimation of the figure of merit. By taking the averaged electrical resistivity ρ of the whole CoFeB/IrMn thin film as $220 \mu\Omega \text{ cm}$ [*Appl. Phys. Lett.* 109, 222401 (2016)] and the thermal conductivity $\kappa = 9 \text{ W m}^{-1}\text{K}^{-1}$ [*Adv. Funct. Mater.* 26, 5507-5514 (2016)], we estimated the figure of merit [$zT = S^2 T / (\rho \kappa)$] for the sample MgO (2 nm)/CoFeB (0.9 nm)/IrMn (3.1 nm) [$S = 392 \mu\text{V K}^{-1}$] at the critical temperature (290 K) to be 2.2. This is quite a large value but this is a rough estimate and it does not take into account the heat flow in the substrate.

In response to the reviewer and considering some readers are likely to have the same question, we added this calculation in the supplementary information and one sentence in the main text referring to it as

“The figure of merit is estimated in Supplementary Section IV.”

4) The cited literature is somehow incomplete. The authors cite many papers from spin caloritronics, which has not much to do with their Seebeck measurements. The authors have probably overlooked some important papers from long ago such as:

M. Ausloos, *Solid State Communications*, Volume 21, Issue 4, January 1977, Pages 373-375

A. Fote et al., *PHYSICAL REVIEW B VOLUME 8, NUMBER 5, PAGE 2099* (Sept. 1973)

Also younger papers such as H. Takaki et al., *Appl. Phys. Lett.* 110, 072107 (2017); <https://doi.org/10.1063/1.4976574> are missing.

I cannot give a full list of adequate references, but the authors should at least cite some work on thermoelectrics with antiferromagnets.

Response:

We thank Reviewer #2 for pointing out these references. We also realised that we should cite more work on thermoelectrics. The previous manuscript was prepared in a form of the journal of Nature, which is limited to 30 references. Therefore, in addition to the references suggested by this Reviewer and the others, we have added more references on both thermoelectrics for applications and thermoelectrics with antiferromagnets such as:

Dresselhaus, M. S. et al. New directions for low-dimensional thermoelectric materials. Adv. Mater. 19, 1043-1053 (2007).

Heremans, J. P. et al. Enhancement of thermoelectric efficiency in PbTe by distortion of the electronic density of states. Science 321, 554-557 (2008).

Tsujii, N. & Mori, T. High Thermoelectric power factor in a carrier-doped magnetic semiconductor CuFeS₂. Appl. Phys. Express 6, 043001 (2013).

Zhao, L.-D. et al. Ultralow thermal conductivity and high thermoelectric figure of merit in SnSe crystals. Nature 508, 373-377 (2014).

Takaki, H. et al. First-principles calculations of Seebeck coefficients in a magnetic semiconductor CuFeS₂. Appl. Phys. Lett. 110, 072107 (2017).

Zhu, T. et al. Compromise and synergy in high-efficiency thermoelectric materials. Adv. Mater. 29, 1605884 (2017).

Tewari, G. C., Tripathi, T. S., Yamauchi, H. & Karppinen, M. Thermoelectric properties of layered antiferromagnetic CuCrSe₂. Mater. Chem. Phys. 145, 156-161 (2014).

Ikhlas, M. et al. Large anomalous Nernst effect at room temperature in a chiral antiferromagnet. Nat. Phys. 13, 1085-1090 (2017).

Narita, H. et al. Anomalous Nernst effect in a microfabricated thermoelectric element made of chiral antiferromagnet Mn₃Sn. Appl. Phys. Lett. 111, 202404 (2017).

Honda, F. et al. Magnetotransport as a probe of phase transformations in metallic antiferromagnets: The case of UIrSi₃. Phys. Rev. B 100, 014401 (2019).

5)The paper thus reports a potentially very valuable result, but it needs improvements and some additional work to be acceptable for Nat.Com..

One minor issue is the following: The figures 2 and sup-7 show the Seebeck voltage of the meandering sample. Why is this voltage at maximum around 45 mV in figure 2 and 400 mV in figure sup-7?

Response:

We thank the Reviewer for pointing this possible confusion. Sorry for the misunderstanding. Actually, the results shown in the inset of Fig. 2C were measured at 295 K with heating current of only 5 mA on sample with $t_{\text{AFM}} = 2.8$ nm, while the results in sub-fig. 7 were obtained at 330 K with heating current up to 20 mA on sample with $t_{\text{AFM}} = 2.8$ nm. The voltage obtained with 5 mA heating current in sub-fig. 7 was comparable with the voltage obtained in Fig. 2C at 330 K. We have now made this clear in the captions.

Reviewer #3:

The authors present a detailed thermopower study of IrMn/CoFeB thermopiles, and report a peak in the Seebeck coefficient at what appears to be the critical temperature of the IrMn thin film. A record value is observed at that temperature for a specific thickness. The results are overall rather interesting, and could be published in Nature Communications if several primary and secondary aspects are satisfactorily addressed.

Primary:

1)Line 90 :”This value is substantially larger than most metallic structures and even larger than some well-known thermoelectric materials, such as Bi₂Te₃ (-160 $\mu\text{V K}^{-1}$).” Please provide more refs/background to substantiate this claim.

Response:

We are grateful to Reviewer #3 for bringing out this point. As shown in the response to Reviewer #1, we have added more references and background into reference to substantiate this claim. Following the suggestion of Reviewer #1, we added a figure (Fig. 4) to compare our results with others. We have also added references, such as:

Gallo, C. F., Chandrasekhar, B. S. & Gallo, C. F. Transport properties of bismuth single crystals. J. Appl. Phys. 34, 144 (1963).

Scheele, M. et al. Synthesis and thermoelectric characterization of Bi₂Te₃ nanoparticles. Adv. Funct. Mater. 9, 3476-3483 (2009).

Hence, we updated the sentence in the main text to refer to these additional references as:

“This value is substantially larger than most metallic structures and even larger than some well-known thermoelectric materials, such as Bismuth ($-70 \mu V K^{-1}$)⁴² and Bi_2Te_3 ($-160 \mu V K^{-1}$)^{43,44}.”

2) In Fig. 3, the link between the Seebeck coeff. Peak and the IrMn AFM critical temperature is experimentally tenuous at present, with only a few magnetic susceptibility and XLMD measurements to overlap with the temperature of the Seebeck coeff. Peak. Please perform additional magnetic susceptibility measurements at other thicknesses in order to conform to experiment and to the model (black line of Fig. 3)

Response:

Thanks to the suggestion of Reviewer #3, which would be very helpful for making our finding more solid. Thus, we have performed additional magnetic susceptibility measurements on the samples with $t_{AFM} = 2.5$ nm and 3.0 nm thickness IrMn. The raw data of the measurements have been added into Supplementary Fig. 9, from which one could extract the critical temperatures to be about 285 K for $t_{AFM} = 2.5$ nm and 290 K for $t_{AFM} = 3.0$ nm. We have now added these critical temperature points into Fig. 3a.

For the XMLD data, we have added the data for $t_{AFM} = 3.2$ nm. After re-analyze the data as stated in the response to Reviewer #1, we have updated the three data points in Fig. 3a for $t_{AFM} = 2.0$ nm, 3.2 nm and 4.0 nm, accordingly.

We have also sharpened the text and the figure caption, so the reader can see that all of the data are consistent with a theoretical model.

3)Line 194: “To fully understand the CoFeB layer influence requires 195 further investigations and is beyond the scope of this work.” This record thermopower is found at the (supposed) IrMn critical temperature. In a commercial device, the IrMn would not be designed such that the device operates at the IrMn critical temperature, because doing so would not allow for magnetic pinning of one of the FM layers. Thus, for the main claim of the paper to stand, this requires that this IrMn layer not play a role in the magnetostatics of the film. This 1) weakens the importance of the paper’s main claim and 2) requires that the IrMn be placed adjacent to non-magnetic layers so as to destabilize (electrically/magnetically) the system. Please discuss this in the context of Ref. 17, which associates the thermopower in these systems with exchange bias.

Response:

We greatly appreciate Reviewer #3 for pointing out that our text needed clarification on the role of exchange biasing.

In Ref. 17 (now Ref. 33), the exchange biasing is dependent on IrMn thickness, allowing the authors to create a thermopile with branches of two different IrMn thicknesses. Please note that in that paper, the thickness is greater than 10 nm, well above the thicknesses considered in our manuscript. We have changed the text to make clear that the exchange biasing is very small (about 1 Oe) and that the Seebeck hardly changes when the magnetization is flipped.

“The exchange bias appears to be rather small (around 1 Oe), which is due to the thicknesses used in this work are smaller than the minimal thickness (~ 10 nm) needed for exchange bias as used in Ref. 33.”

[33] Kim, D. J. et al. Utilization of the antiferromagnetic IrMn electrode in spin thermoelectric devices and their beneficial hybrid for thermopiles. *Adv. Funct. Mater.* **26**, 5507-5514 (2016).

4)The authors do not really discuss why the Seebeck coeff. Maximum is highest at a precise thickness. This will be required to publish in Nat. Comm.

Response:

We are grateful for Reviewer #3 pointing out that we may have generated a confusion. The peak in the Seebeck versus thickness data is only apparent. Its actual cause is the strong temperature dependence around the Néel temperature combined with the thickness dependence of the Néel temperature.

In order to avoid this confusion, we have changed the title of Fig. 1, which now reads

“Enhanced Seebeck effect when the thickness-dependent Néel temperature matches the measurement temperature.”

We also state in the text:

“The apparent thickness dependence arises from the thickness dependence of the Néel temperature, as discussed below. The Seebeck coefficient S is optimal at room temperature when the thickness is 3.1 nm.”

and put “RT (295 K)” in Fig. 1c for clarification.

In other words, the reason why the Seebeck coefficient is highest at 3.1 nm IrMn is that the critical temperature in this sample is very close to room temperature. For the other thicknesses, if the thickness of IrMn is below 3.1 nm, the critical temperature of peak position in the temperature-dependent measurements will move to lower temperature range, if the thickness is bigger than 3.1 nm, the critical temperature of peak position in the temperature-dependent measurements will move to higher temperature range than room temperature. Therefore, the Seebeck coefficient will decrease if the thickness of IrMn is smaller or bigger than 3.1 nm.

5)The theoretical description is rather vague.

Response:

Reviewer #3 and Reviewer #1 made similar suggestions. We thank both for making realize that some of the theoretical discussion should not be relegated to the supplementary information, but included in the manuscript itself.

Secondary

6)Line 81 “with minimal intermixing”: that doesn’t seem so based on Fig. S1. Also, are each panel of Fig. S1 at the same position? One gets the impression that Fe and Mg (bright streaks) are right of Mn despite the total color scale map showing otherwise. Clearly, Fig. S1 is confusing as is.

Response:

We greatly appreciate Reviewer #3 for pointing this out. We have simply removed this statement.

Mechanics:

7)Ordering of figure description out of order with Figure assembly. For example, it isn’t natural to describe the record Seebeck value before describing the thermopile.

Response:

We thank Reviewer #3 for carefully reading and great suggestions. Actually, the Seebeck coefficient as a function of thickness at room temperature in Fig. 1c is measured using a single rectangular bar which is illustrated by the schematics in Fig. 1a. The schematics in Fig 2a shows instead a meander, the thermopile with which we performed the temperature dependence measurements.

8) Please check manuscript and SI consistency, e.g. : “Based on the single rectangular bar device in Supplementary Fig. 3”, even though Fig S3 is about how to measure the thermal gradient using data extracted from the thermal camera.

Response:

We thank Reviewer #3 for his/her careful reading and finding this typo. We have now made the correction.

“Based on the single rectangular bar device in Fig. 1a in the main text.”

9) Please indicate IrMn thickness in all plots of all relevant Figures, including Fig. S4.

- Fig 1c: indicate the temperature in the graph

Response:

We are grateful to Reviewer #3 for bringing out these points. We have put the IrMn thicknesses in all plots of relevant figures and added “RT (295 K)” in Fig. 1c.

10) The table caption is quite minimal. For example, is N the Nernst coefficient? It isn't defined at the first appearance in the text?

Response:

We thank Reviewer #3 for his/her careful reading and pointing this out. Since Reviewer #1 suggested to remove the table, the table has been removed altogether.

English

11) Here are a few examples of minor English improvements that are needed throughout the text. Please improve throughout through a careful rereading. Line 48: junctions - > junction

Response:

We greatly appreciate Reviewer #3 for his/her great help and suggestions on helping improve the English of the manuscript. We fully agree and have modified this accordingly.

We have also gone through one careful reading of the entire manuscript and corrected the English wherever necessary.

12) Line 58: We report a record Seebeck coefficient of up to $392 \mu\text{V K}^{-1}$, tunable over a wide 59 temperature range, including room temperature, by the thickness of the antiferromagnetic IrMn layer. -> improve English: We report a record Seebeck coefficient of up to $392 \mu\text{V K}^{-1}$ that is tunable over a wide 59 temperature range, including room temperature, using the thickness of the antiferromagnetic IrMn layer.

Response:

We thank Reviewer #3 for the helpful suggestion on polish the English of the manuscript to be more scientifically sound. We fully agree and have modified this accordingly. This passage now reads

“Here, we investigate the thermoelectric response of IrMn/CoFeB near the IrMn critical temperature. We report a record Seebeck coefficient of $392 (\pm 13) \mu\text{V K}^{-1}$ at room temperature and a strong temperature dependence around the phase transition of IrMn, which depends on thickness. The highest value was $1.1 \pm 0.1 \text{ mV K}^{-1}$ at 270K for a thickness of 2.5 nm of IrMn.”

13) “Line 62: “The peak position of the 62 thermopower” -> you mean the temperature at which the peak occurs, please improve. Critical temperature of IrMn : AFM?”

Response:

We thank Reviewer #3 for the helpful suggestion on the uncertainty to be more scientifically sound. We fully agree and have modified this accordingly.

“In our micro-structured thermoelectric devices, our temperature dependent measurements show a sharp peak of the thermopower around the critical temperature (T_{crit}) of the IrMn film. The peak position of the thermopower can be shifted by changing the IrMn thickness, and directly correlates with the T_{crit} characterized by magnetic susceptibility and X-ray magnetic linear dichroism (XMLD) measurements.”

14)SI: “which includes only 2 atoms of IrMn” -> atomic planes

Response:

We thank Reviewer #3 for the helpful suggestion on the description to be more scientifically sound. We fully agree and have modified this accordingly.

“which includes only two atomic planes of IrMn”

15) “which had the highest Seebeck coefficient”

Response:

We thank Reviewer #3 for his/her careful reading and finding this typo and we have now made the correction.

“which had the highest Seebeck coefficient”

16) “The discrepancy may because of different IrMn targets and crystal orientation of the thin film.”

Response:

We thank Reviewer #3 for his/her careful reading and pointing out this grammatical mistake and we have now rewritten this sentence:

“The discrepancy may be induced by different IrMn targets and crystal orientation of the thin film.”

17)Line 86 : “The Seebeck coefficientS, is”

Response:

We thank Reviewer #3 for his/her careful reading and finding this typo and we have now made the correction as

“The Seebeck coefficient is”

Reviewers' comments:

Reviewer #1 (Remarks to the Author):

Upon re-reading the manuscript from Tu et al, I find the work much improved compared to the previous version. The comments I have on the manuscript are arranged against the major points raised in the previous version.

- Firstly, it should be noted that the issues with the presentation and analysis of the XMLD section of the paper have been addressed completely. The presentation and description of the data is much clearer and makes the claims easier to substantiate. The description in the main body of the text helps to then convey the understanding of the importance of this result in a much more effective way than in the previous version.
- Also much improved in this version is the discussion of the scientific context of the result with regards to the current state of the art. The addition of Fig. 4 in particular is an effective way of conveying the scale of the achievement.
- I also appreciate the introduction of the discussion regarding the anomalous Nernst effect and its relationship to the Seebeck effect. The description is clear and helps the understanding of the importance of the scientific result of the paper. The discussion regarding the influence of the dimensionality of the fluctuations also helps make a case for the exceptionally large Seebeck coefficient. However, I do have concerns regarding this discussion which are outlined below.
- The introduction of the error analysis into the manuscript, particularly in Fig 1 and the quoted values, is most welcome. However, I have issues with the presentation of the errors which I list explicitly in a separate section.

Issues with the errors:

- Line 38, 63: When quoting error bars on values it is necessary to present only one significant figure due to the statistical uncertainty of performing a single measurement. For a full explanation of this see Taylor, Introduction to error analysis: The Study of Uncertainties in Physical Measurements, or similar. I realise that this is very pedantic but I hope the authors understand why this is important and see fit to alter the quoted value.
- Fig 3a. Please add error bars to the data points here.
- Line 214: please add an error bar to the Seebeck coefficient of the new control sample.

Aside from the error analysis, the only outstanding issue I have with the manuscript in its current form is the discussion of the origin of the large Seebeck coefficient in line 199 – 205. Below I've outlined my suggestions for improvement:

- It's not clear to me why there would be a transition from metallic to localized behaviour in the IrMn here given the lack of a corresponding change in the resistivity trace in Supp. Fig. 2b? Perhaps an explanation where this idea comes from would be beneficial.
- I find the sentence on line 201 beginning 'Thus whether the theory...' difficult to read. The word thus is used incorrectly in this sentence.
- In the supplementary section in which the theoretical considerations are discussed, the contributions of charge and heat currents are discussed, but spin-currents are not. Given that this system is an AFM in contact with an FM, and the large Seebeck coefficient comes when the two are in contact and not the individual layers (new Fig 4), it seems that the spin-current contribution to the Seebeck coefficient may not be negligible in this system.
- So my question is this: Is there a reason why the contribution of magnons/spin-currents are not considered here? The authors have already stated that increased spin fluctuations in the IrMn around the Neel temperature contribute to the peak in the Seebeck coefficient. So therefore, why, as you have a FM in contact with the AFM, could the large signal not come from a spin-current entering the AF layer?
- The contribution of magnons to the Seebeck effect in FM/NM bilayers has been demonstrated previously (Rezende et al, Phys Rev B 89, 014416, 2014). The transfer of spin current through the AF layer would be increased around the Neel temperature and may contribute to the large signal. Such behaviour is not dissimilar to that presented in the Frangou paper already cited in the

manuscript.

- Please add a brief discussion regarding the role of the magnon/spin transport in the system and its possible contribution to the large Seebeck coefficient.

To summarize, on second reading the manuscript is much improved and the arguments have a more robust scientific validity. All of the issues I had previously have been addressed and those that have arisen in the new version of the manuscript can be simply rectified. Therefore, should the authors address the issues regarding the error bars raised in this review and add a discussion regarding the role of spin-currents in the system then I believe the article would be suitable for publication in Nature Communications.

Reviewer #2 (Remarks to the Author):

The authors did a good revision of their manuscript. I think the manuscript is now acceptable for publication in Nature Communications. I just have two suggestions that could further improve the manuscript:

1. At my point of view, they discovered an important point during doing additional work: by removing the CoFeB, the Seebeck voltage drops at the peak position by a factor of seven. This indicates, that the presence of a ferromagnet at the interface enhances the mechanism that lead to the Seebeck voltage. This could be addressed a bit more in the manuscript.
2. I still insist, that the Nernst voltage is not produced in the IrMn but stems from the CoFeB. Thus the discussion that relates the Nernst voltage to overall (mean) values of physical parameters (such as σ) of the complete stack are probably misleading. The role of the interface of the CoFeB to the IrMn is, however, also important for the Nernst effect. Thus, the scattering mechanisms that contribute to the Nernst voltage are probably influenced by this interface.

Reviewer #3 (Remarks to the Author):

The authors have satisfactorily addressed my previous report, as well as those of the other reviewers, such that I can anticipate recommending publication once the points below are satisfactorily addressed, to the best of the authors' abilities and given the present state-of-the-art.

- In their response, the authors satisfactorily address the link between the transition temperature of the IrMn thin film and the maximum Seebeck coefficient S measured at RT for a film of appropriate thickness. The demonstration would be strengthened if measurements of S for at least one additional (thickness, temperature) couple were shown in the main text. I therefore recommend that the data of Fig. S8 be included in the main text, and discussed (this is lacking presently). Why is S peaking at a specific (thickness, temperature) couple?

- The newly introduced discussion on the impact of the CoFeB layer on the effect is somewhat vague but at the state of the art. I do wonder if, aside from possible explanations in terms of DMI and Rashba terms, one should also consider how the proximity of the CoFeB layer to the IrMn layer, and the resulting effective internal magnetic field, will influence the latter's spin fluctuations, which the authors state are central (line 181) to the large Seebeck coefficient at the IrMn phase transition.

- Since the paper considers magnetic tunnel junctions as a possible implementation of their discovery, it would be useful background to cite the recent work of Katcko et al Commun. Phys. 2, 116 on generating electricity in this kind of spintronic device using quantum thermodynamics. In addition, that paper's discussion on effective magnetic fields and spintronic anisotropy might also help in addressing the previous point.

- The authors have not fully addressed my previous primary point 3. To avoid any confusion, the

manuscript should explicitly mention that the magnetic pinning and thermoelectric functionalities of the IrMn layer cannot be combined within a spintronic device using only one IrMn layer. To get both properties would require two layers within the MTJ stack.

- Fig 2c: please indicate the thickness in the Figure. This helps the reader to quickly assess the dependent parameters in the experiment, here namely how the critical temperature is IrMn-thickness dependent.

Minor English

Despite my previous entreaty to carefully look at the English, several mistakes remain in the text:

- "an IrMn-based » -> « IrMn-based »

- "is the magnetic tunnel junctions (MTJ) » -> « is the magnetic tunnel junction (MTJ) »

Please comb the text carefully for additional ones.

Response to referee reports

Reviewer #1:

Upon re-reading the manuscript from Tu et al, I find the work much improved compared to the previous version. The comments I have on the manuscript are arranged against the major points raised in the previous version.

-Firstly, it should be noted that the issues with the presentation and analysis of the XMLD section of the paper have been addressed completely. The presentation and description of the data is much clearer and makes the claims easier to substantiate. The description in the main body of the text helps to then convey the understanding of the importance of this result in a much more effective way than in the previous version.

-Also much improved in this version is the discussion of the scientific context of the result with regards to the current state of the art. The addition of Fig. 4 in particular is an effective way of conveying the scale of the achievement.

-I also appreciate the introduction of the discussion regarding the anomalous Nernst effect and its relationship to the Seebeck effect. The description is clear and helps the understanding of the importance of the scientific result of the paper. The discussion regarding the influence of the dimensionality of the fluctuations also helps make a case for the exceptionally large Seebeck coefficient. However, I do have concerns regarding this discussion which are outlined below.

-The introduction of the error analysis into the manuscript, particularly in Fig 1 and the quoted values, is most welcome. However, I have issues with the presentation of the errors which I list explicitly in a separate section.

Response:

We thank Reviewer #1 for his/her appreciation of our efforts and improvement made in the first revision of the manuscript, e.g. presentation of the XMLD data and the comparison with the current state of the art, which benefited from the reviewer's constructive comments in the previous review process. In the following, we have carefully considered his/her further suggestions.

Issues with the errors:

- Line 38, 63: When quoting error bars on values it is necessary to present only one significant figure due to the statistical uncertainty of performing a single measurement. For a full explanation of this see Taylor, Introduction to error analysis: The Study of Uncertainties in Physical Measurements, or similar. I realise that this is very pedantic but I hope the authors understand why this is important and see fit to alter the quoted value.

Response:

We thank Reviewer #1 for pointing out that only one significant figure should be presented in the error analysis, which is not pedantic but absolutely the right thing to do in scientific analysis. We therefore have updated/corrected the values quoting error bars as e.g. $390 (\pm 10) \mu V K^{-1}$ throughout the main text and the supplementary information.

-Fig 3a. Please add error bars to the data points here.

Response:

We have now added the error bars in Fig. 3a.

-Line 214: please add an error bar to the Seebeck coefficient of the new control sample.

Response:

The error bar has been added to the Seebeck coefficient as $82 (\pm 3) \mu V K^{-1}$.

Aside from the error analysis, the only outstanding issue I have with the manuscript in its current form is the discussion of the origin of the large Seebeck coefficient in line 199 – 205. Below I've outlined my suggestions for improvement:

-It's not clear to me why there would be a transition from metallic to localized behaviour in the IrMn here given the lack of a corresponding change in the resistivity trace in Supp. Fig. 2b? Perhaps an explanation where this idea comes from would be beneficial.

Response:

We agree with the referee that no tendency towards localisation is visible in our transport measurements. We note that the idea we expressed arises from purely theoretical considerations in that for weakly interacting electrons, the combination of disorder and crossover to two-dimensionality should eventually lead to weak, and finally strong localisation. This tendency will depend on energy. Any asymmetry above and below the Fermi energy will contribute to the thermopower even in a precursor regime, where there is not yet a true metal-insulator transition, as would be visible in transport. As this idea is based on theoretical expectations rather than experimental evidence, it is probably not appropriate to include in this paper. We had thought to include such a remark in the hope that it may inspire future investigations but, on balance, it is probably better to omit it as it, as an unnecessary distraction from the experimental results. Even as a theoretical idea, it requires much more explanation: the cross-over from three to two dimensions is not simple, even for non-interacting electrons and it is not clear that a non-interacting picture is sufficient. We have therefore suppressed the sentence mentioning localisation and hope to return to the issue elsewhere.

-I find the sentence on line 201 beginning 'Thus whether the theory...' difficult to read. The word thus is used incorrectly in this sentence.

Response:

We have now removed the word "Thus".

-In the supplementary section in which the theoretical considerations are discussed, the contributions of charge and heat currents are discussed, but spin-currents are not. Given that this system is an AFM in contact with an FM, and the large Seebeck coefficient comes when the two are in contact and not the individual layers (new Fig 4), it seems that the spin-current contribution to the Seebeck coefficient may not be negligible in this system.

-So my question is this: Is there a reason why the contribution of magnons/spin-currents are not considered here? The authors have already stated that increased spin fluctuations in the IrMn around the Neel temperature contribute to the peak in the Seebeck coefficient. So therefore, why, as you have a FM in contact with the AFM, could the large signal not come from a spin-current entering the AF layer?

-The contribution of magnons to the Seebeck effect in FM/NM bilayers has been demonstrated previously (Rezende et al, Phys Rev B 89, 014416, 2014). The transfer of spin current through the AF layer would be increased around the Neel temperature and may contribute to the large signal. Such behaviour is not dissimilar to that presented in the Frangou paper already cited in the manuscript.

-Please add a brief discussion regarding the role of the magnon/spin transport in the system and its possible contribution to the large Seebeck coefficient.

Response:

We thank Reviewer #1 for this helpful comment. This is indeed a very insightful question and suggestion. As a ferromagnet/antiferromagnet system is investigated, the magnon/spin-transport may reasonably contribute to the spin Seebeck effect. We have carefully looked into the helpful theoretical work of Rezende *et al.* pointed out by the reviewer. Nevertheless, the magnons give rise to the spin Seebeck effect, but not the Seebeck effect, since the electric voltage due to the spin Seebeck effect is perpendicular to the temperature gradient like the Nernst effect. We also searched in the literature for some experimental works. We realize that the magnon/spin-transport mediated spin Seebeck effect in antiferromagnet-based systems can also be enhanced around its Néel temperature, but the amplitude is typically in the range of hundreds of nV to a few μV (references [28,29,31,33]). The large Seebeck voltage detected in this work can reach hundreds of μV and even mV , approximately 1'000 times larger than the output voltage of the magnon-mediated spin Seebeck effect. Although we do not think the magnon-mediated spin Seebeck effect itself can account for the large Seebeck coefficient observed in this work, we completely agree with the reviewer that the contribution from magnon/spin-current need to be discussed. We have therefore added a new sentence including the theoretical reference mentioned by the reviewer together with a very recent experimental one as

“...Furthermore, the magnon spin current at the interface of IrMn/CoFeB can contribute to the spin Seebeck effect as studied previously^{28,29,31,33,66,67}, but the electric voltage due to this effect is perpendicular to the temperature gradient and typically three orders of magnitude smaller than the Seebeck voltage detected in this work.”

[66] Rezende, S. M. et al. Magnon spin-current theory for the longitudinal spin-Seebeck effect. *Phys. Rev. B* **89**, 014416 (2014).

[67] Li, J. et al. Spin current from sub-terahertz-generated antiferromagnetic magnons. *Nature* **578** 70-74 (2020).

More detailed discussions are also added in the supplementary information.

To summarize, on second reading the manuscript is much improved and the arguments have a more robust scientific validity. All of the issues I had previously have been addressed and those that have arisen in the new version of the manuscript can be simply rectified. Therefore, should the authors address the issues regarding the error bars raised in this review and add a discussion regarding the role of spin-currents in the system then I believe the article would be suitable for publication in Nature Communications.

Response:

We have followed the helpful suggestions of Reviewer #1 to improve further the presentation of error analysis and to add some discussions about the potential contribution from magnons at IrMn/CoFeB to the enhanced Seebeck effect. We finally appreciate his/her considering that **our work would be suitable for publication in Nature Communications.**

Reviewer #2:

The authors did a good revision of their manuscript. I think the manuscript is now acceptable for publication in Nature Communications. I just have two suggestions that could further improve the manuscript:

1. At my point of view, they discovered an important point during doing additional work: by removing the CoFeB, the Seebeck voltage drops at the peak position by a factor of seven. This indicates, that the presence of a ferromagnet at the interface enhances the mechanism that lead to the Seebeck voltage. This could be addressed a bit more in the manuscript.

Response:

We thank Reviewer #2 for his/her full support by stating that **our manuscript is now acceptable for publication in Nature Communications.**

As also suggested by Reviewers #1 and #3, we have now added a few more sentences to discuss the important role of the ferromagnet/antiferromagnet interface in terms of the potential contribution from magnon spin current to the Seebeck effect and effective internal magnetic field generated by the magnetic proximity effect. Please see also the replies to Reviewers #1 and #3 for more details.

2. I still insist, that the Nernst voltage is not produced in the IrMn but stems from the CoFeB. Thus the discussion that relates the Nernst voltage to overall (mean) values of physical parameters (such as sigma) of the complete stack are probably misleading. The role of the interface of the CoFeB to the IrMn is, however, also important for the Nernst effect. Thus, the scattering mechanisms that contribute to the Nernst voltage are probably influenced by this interface.

Response:

We agree with the reviewer that the anomalous Nernst effect should stem from the ferromagnetic layer that is CoFeB. Therefore, we have now replaced the argument on overall sigma to a newly added sentence concerning the interface of CoFeB/IrMn as

“...That the two are correlated implies that the ferromagnet/antiferromagnet interface plays an important role not only in the Seebeck effect but also in the anomalous Nernst effect.”

Reviewer #3:

The authors have satisfactorily addressed my previous report, as well as those of the other reviewers, such that I can anticipate recommending publication once the points below are satisfactorily addressed, to the best of the authors' abilities and given the present state-of-the-art.

Response:

We are grateful to Reviewer #3 for considering that our revision satisfactorily addressed previous comments/suggestions and recommending publication once the further points are satisfactorily addressed. We have therefore made efforts to further improve the manuscript according to the constructive suggestions of the reviewer as the followings.

- In their response, the authors satisfactorily address the link between the transition temperature of the IrMn thin film and the maximum Seebeck coefficient S measured at RT for a film of appropriate thickness. The demonstration would be strengthened if measurements of S for at least one additional (thickness, temperature) couple were shown in the main text. I therefore recommend that the data of Fig. S8 be included in the main text, and discussed (this is lacking presently). Why is S peaking at a specific (thickness, temperature) couple?

Response:

This is a very important and helpful suggestion. We have followed the advice of the reviewer and added a new Fig. 3 together with some discussions on this figure as

“The data points in Fig. 3 are Seebeck coefficients at 270 K obtained from the temperature-dependent measurements on samples with different thicknesses (Supplementary Fig. 6). The Seebeck coefficient peaks at the sample with $t_{AFM}=2.5$ nm reaching 1.1 ± 0.1 mV K⁻¹ at 270 K. Hence, the Seebeck coefficient peak position depends jointly on the measurement temperature and the IrMn thickness.”

- The newly introduced discussion on the impact of the CoFeB layer on the effect is somewhat vague but at the state of the art. I do wonder if, aside from possible explanations in terms of DMI and Rashba terms, one should also consider how the proximity of the CoFeB layer to the IrMn layer, and the resulting effective internal magnetic field, will influence the latter's spin fluctuations, which the authors state are central (line 181) to the large Seebeck coefficient at the IrMn phase transition.

Response:

We thank Reviewer #3 for pointing out another potential cause for the enhanced Seebeck coefficient induced by the CoFeB layer. We have accordingly added two sentences including one reference on the magnetic proximity effect as

“...The CoFeB layer may also induce the magnetic proximity effect⁶⁴, which generates an effective internal magnetic field in the adjacent IrMn layer. This effective magnetic field may influence the spin fluctuations at the IrMn/CoFeB interface and eventually enhance the Seebeck coefficient.”

[65] Huang, S. Y. et al. Transport magnetic proximity effects in platinum. *Phys. Rev. Lett.* **109**, 107204 (2012).

- Since the paper considers magnetic tunnel junctions as a possible implementation of their discovery, it would be useful background to cite the recent work of Katcko et al *Commun. Phys.* 2, 116 on generating electricity in this kind of spintronic device using quantum thermodynamics. In addition, that paper's discussion on effective magnetic fields and spintronic anisotropy might also help in addressing the previous point.

Response:

This recent reference is indeed very helpful and we have now added it when mentioning the magnetic tunnel junctions.

[23] Katcko, K. et al. Spin-driven electrical power generation at room temperature. *Commun. Phys.* 2, 116 (2019).

- The authors have not fully addressed my previous primary point 3. To avoid any confusion, the manuscript should explicitly mention that the magnetic pinning and thermoelectric functionalities of the IrMn layer cannot be combined within a spintronic device using only one IrMn layer. To get both properties would require two layers within the MTJ stack.

Response:

We completely agree and have now added one sentence to make an explicit statement at the very end of the manuscript as

“In a spintronic device, the IrMn layer for thermoelectric functionalities need to differ from the one for the magnetic pinning due to dissimilar active thicknesses.”

- Fig 2c: please indicate the thickness in the Figure. This helps the reader to quickly assess the dependent parameters in the experiment, here namely how the critical temperature is IrMn-thickness dependent.

Response:

We have followed the reviewer’s helpful suggestion to add the thickness “ $t_{AFM} = 2.8 \text{ nm}$ ” in Fig. 2c.

Minor English

Despite my previous entreaty to carefully look at the English, several mistakes remain in the text:

- “an IrMn-based » -> « IrMn-based »

- “is the magnetic tunnel junctions (MTJ) » -> « is the magnetic tunnel junction (MTJ) »

Please comb the text carefully for additional ones.

Response:

We thank Reviewer #3 for carefully reading our manuscript and kindly pointing out these typos in the English. We have now rectified these typos and additional ones when going once more through the main text and the supplementary information.

Reviewers' comments:

Reviewer #1 (Remarks to the Author):

Upon a third reading of the manuscript I believe that all of the issues raised in past reviews have been satisfactorily addressed. As a result of the changes made by the authors I find the manuscript much improved compared to the initial submission. As such, I can now recommend this work for publication in Nature Communications.

Reviewer #2 (Remarks to the Author):

The authors have now improved the manuscript in a way that makes it acceptable for Nature Communications. Although there are -as usual- still open issues (such as the role of the IrMn/CoFeB interface and the source of the large Seebeck coefficient), I can now recommend publication as is.

Reviewer #3 (Remarks to the Author):

In the latest review round, the authors have once more improved their manuscript in a mostly satisfactory fashion. The only remaining point prior to recommending publication, which wasn't addressed in the previous round, is:

- Why is S peaking at a specific (thickness, temperature) couple?

Please provide your best answer in the manuscript.

Minor aspects:

The title reads: "Record thermopower found in an IrMn-based spintronic device". The title seems misleading since the effect wasn't achieved on a spintronic device, but rather part of a spintronic stack from which a spintronic device could be made. The spintronic device isn't operated in the manuscript. I recommend changing the title to "Record thermopower found in an IrMn-based spintronic stack". I leave this to editorial discretion.

"In a spintronic device, the 235 IrMn layer for thermoelectric functionalities need to differ from the one for the magnetic pinning 236 due to dissimilar active thicknesses."

> 'needs' as a minimum correction, but I recommend:

"In a spintronic device, the different thicknesses of IrMn required to achieve the thermoelectric and magnetic pinning properties at a given temperature point can be solved by integrating two IrMn layers in the stack."

Response to referee reports

Reviewer #1:

Upon a third reading of the manuscript I believe that all of the issues raised in past reviews have been satisfactorily addressed. As a result of the changes made by the authors I find the manuscript much improved compared to the initial submission. As such, I can now recommend this work for publication in Nature Communications.

Reviewer #2:

The authors have now improved the manuscript in a way that makes it acceptable for Nature Communications. Although there are -as usual- still open issues (such as the role of the IrMn/CoFeB interface and the source of the large Seebeck coefficient), I can now recommend publication as is.

Response:

We thank Reviewers #1 and #2 for their full support and recommendation for publication in Nature Communications.

Reviewer #3:

In the latest review round, the authors have once more improved their manuscript in a mostly satisfactory fashion. The only remaining point prior to recommending publication, which wasn't addressed in the previous round, is:

- Why is S peaking at a specific (thickness, temperature) couple?

Please provide your best answer in the manuscript.

Response:

We have now seriously considered the constructive suggestion from Reviewer #3 and have therefore added a few sentences in the main text to further explain the reason why the Seebeck coefficient peaks at particular (thickness, temperature) couple as:

“The reason for S peaking at a specific couple of the temperature and thickness is two-fold: 1) The temperature-dependent Seebeck coefficient peaks around the IrMn Néel temperature T_{crit} (Fig. 2); 2) T_{crit} is strongly dependent on the thickness of IrMn²⁵.”

Minor aspects:

The title reads: “Record thermopower found in an IrMn-based spintronic device”. The title seems misleading since the effect wasn't achieved on a spintronic device, but rather part of a spintronic stack from which a spintronic device could be made. The spintronic device isn't operated in the manuscript. I recommend changing the title to “Record thermopower found in an IrMn-based spintronic stack”. I leave this to editorial discretion.

Response:

Indeed, Reviewer #3 has reasons and we have changed accordingly the wording from “device” to “stack” in the title.

“In a spintronic device, the 235 IrMn layer for thermoelectric functionalities need to differ from the one for the magnetic pinning 236 due to dissimilar active thicknesses.”

> ‘needs’ as a minimum correction, but I recommend:

“In a spintronic device, the different thicknesses of IrMn required to achieve the thermoelectric and magnetic pinning properties at a given temperature point can be solved by integrating two IrMn layers in the stack.”

Response:

We thank Reviewer #3 for the helpful improvement on this sentence in the discussion. We fully agree and have modified the text with this suggested expression.

REVIEWERS' COMMENTS:

Reviewer #3 (Remarks to the Author):

The remaining points have been addressed satisfactorily, such that I now unreservedly recommend publication.

Response to referee reports

Reviewer #3:

The remaining points have been addressed satisfactorily, such that I now unreservedly recommend publication.

Response:

We thank Reviewers #3 for his/her full support and recommendation for publication in *Nature Communications*.